# Role for the shelterin protein TRF2 in human herpesvirus 6A/B chromosomal integration

**Shella Gilbert-Girard**[1⊚], **Annie Gravel**[1⊚], **Vanessa Collin**[1], **Darren J. Wight**[2], **Benedikt B. Kaufer**[2], **Eros Lazzerini-Denchi**[3], **Louis Flamand**[1,4]\*

**1** Division of Infectious Disease and Immunity, CHU de Québec Research Center, Quebec City, Quebec, Canada, **2** Institut für Virologie, Freie Universität Berlin, Berlin, Germany, **3** Laboratory of Genome Integrity, National Cancer Institute, National Institutes of Health, Bethesda, Maryland, United States of America, **4** Department of microbiology, infectious diseases and immunology, Faculty of Medicine, Université Laval, Quebec City, Québec, Canada

⊚ These authors contributed equally to this work.

\* Louis.flamand@crchudequebec.ulaval.ca

**Data Availability Statement:** All relevant data are within the manuscript and its Supporting Information files.

## Abstract

Human herpesviruses 6A and 6B (HHV-6A/B) are unique among human herpesviruses in their ability to integrate their genome into host chromosomes. Viral integration occurs at the ends of chromosomes within the host telomeres. The ends of the HHV-6A/B genomes contain telomeric repeats that facilitate the integration process. Here, we report that productive infections are associated with a massive increase in telomeric sequences of viral origin. The majority of the viral telomeric signals can be detected within viral replication compartments (VRC) that contain the viral DNA processivity factor p41 and the viral immediate-early 2 (IE2) protein. Components of the shelterin protein complex present at telomeres, including TRF1 and TRF2 are also recruited to VRC during infection. Biochemical, immunofluorescence coupled with in situ hybridization and chromatin immunoprecipitation demonstrated the binding of TRF2 to the HHV-6A/B telomeric repeats. In addition, approximately 60% of the viral IE2 protein localize at cellular telomeres during infection. Transient knockdown of TRF2 resulted in greatly reduced (13%) localization of IE2 at cellular telomeres ($p<0.0001$). Lastly, TRF2 knockdown reduced HHV-6A/B integration frequency ($p<0.05$), while no effect was observed on the infection efficiency. Overall, our study identified that HHV-6A/B IE2 localizes to telomeres during infection and highlight the role of TRF2 in HHV-6A/B infection and chromosomal integration.

## Author summary

Human herpesvirus 6A and 6B (HHV-6A/B) are able to integrate their genome into host chromosomes. Viral integration occurs at the ends of chromosomes within the host telomeres, containing hundreds to thousands tandem TTAGGG repeats. The ends of the HHV-6A/B genomes also contain telomeric repeat arrays (15–180 repeats) that play an important role in chromosomal integration; however, their impact on telomere homeostasis remains unknown. In the current study we provide evidence that HHV-6A/B DNA

**Funding:** This work was funded by a Canadian Institutes of Health Research grants (www.cihr-irsc.gc.ca) (MOP_123214 and PJT_156118) awarded to LF and an European Research Council (ERC) (https://erc.europa.eu) grant (Stg 677673) awarded to BBK. The funders had no role in study design, data collection and analysis, decision to publish, or preparation of the manuscript.

**Competing interests:** The authors have declared that no competing interests exist.

replication dramatically increases the number of telomeric repeats in infected cells. We report for the first time that the viral IE2 protein colocalizes with cellular telomeres. In addition, we demonstrate that cellular telomere binding proteins such as TRF1 and TRF2 bind to the viral telomeric repeats during infection. Importantly, we could demonstrate that TRF2 plays an important role in HHV-6A/B integration. Overall, our results highlight the role of shelterin complex proteins during replication and integration of HHV-6A/B into host telomeres.

## Introduction

Human herpesvirus-6A (HHV-6A) and HHV-6B (HHV-6A/B) are two distinct betaherpesviruses with different epidemiological and biological characteristics [1]. HHV-6B is a ubiquitous virus that infects nearly 100% of world population and is the etiological agent of roseola infantum, an infantile febrile illness characterized by high fever and skin rash [2]. HHV-6B is also a concern in hematopoietic stem cell and solid organ transplant recipients with frequent reactivation and medical complications [3]. Pathological and epidemiological data on HHV-6A remain scarce.

The viral genomes of HHV-6A/B are composed of a unique segment of approximately 143 kilo base pair (kbp) flanked at both extremities with identical and directly repeated (DR) termini of approximately 9 kbp each [4, 5]. Each DR contains two TTAGGG telomeric repeat arrays that are required for the integration of the HHV-6A/B genome into human chromosomes [6] and reviewed in [7, 8]. The number of TTAGGG telomeric repeats within each DR ranges from 15 to 180 in clinical isolates [9–12]. HHV-6A/B integration can occur in several distinct chromosomes, and it invariably takes place in the telomeric/sub-telomeric regions of chromosomes [13–16]. When integration occurs in a gamete, the viral genome can be inherited resulting in individuals carrying a copy of the viral genome in every cell, a condition called inherited chromosomally-integrated HHV-6A/B (iciHHV-6A/B) [17]. Approximately 1% of the world population have this condition (reviewed in [8]). Chromosomally-integrated HHV-6A/B are not silent and can spontaneously express viral genes with the U90 and U100 generally being the most abundant in various tissues [18, 19]. Expression of such viral genes is correlated with a greater antibody response in iciHHV-6A/B subjects relative to age and sex matched controls [19]. The consequences of having iciHHV-6A/B are not well defined but a recent study suggests that these individuals are at greater risks of developing angina pectoris [20].

Mammalian chromosome extremities consist in 5–50 kbp of $(TTAGGG)_n$ repeats followed by a single-stranded 3'-overhang of 200 +/-75 TTAGGG nucleotides [21]. With each cell division, the extremities of the chromosomes are not completely replicated due to the end replication problem of the cellular DNA polymerase [22]. As a consequence, the telomeres shorten with every cell division until they reach a minimal length threshold, which triggers DNA damage activation via the ATR (ataxia telangiectasia and Rad3 related) or the ATM (ataxia telangiectasia mutated) pathway, ultimately leading to senescence or cell death (reviewed in [23, 24]). In the absence of telomere elongation processes, such as expression of telomerase or activation of the alternative lengthening of telomere (ALT) pathway, somatic cells are limited in their number of replication cycles.

To prevent activation of DNA damage recognition pathways, telomeres are protected by a six-member protein complex, termed shelterin [23]. The shelterin components TRF1 and TRF2 form homodimers that bind the double-strand TTAGGG repeats and recruit the rest of the complex (TPP1, RAP1, TIN2 and POT1) to chromosome ends [25–27]. The shelterin complex folds telomeric DNA into a secondary structure called the T-Loop, preventing the

recognition of the telomere extremity as a double-strand break (DSB) [28]. TRF2 represses activation of the ATM pathway [29] and plays an essential role in end-to-end chromosome fusions mediated by the non-homologous end-joining (NHEJ) pathway [30, 31]. POT1 binds to the single-strand section of the telomeres and protects the telomeres against activation of the ATR pathway [30, 32–34].

Certain viruses are reported to affect telomeres in different ways. For example, infection by herpes simplex virus type 1 (HSV-1) alters telomere integrity through transcriptional activation of the telomeric noncoding RNA (TERRA), loss of total telomeric DNA, selective degradation of TPP1, reduction of telomere-bound shelterin and accumulation of DNA damage at telomeres [35]. Telomere remodeling is presumed to be required for ICP8-nucleation to form a pre-replication compartment that stimulates HSV-1 replication [35]. The Epstein-Barr virus (EBV) LMP1 protein was reported to downmodulate the expression of TRF1, TRF2 and POT1 shelterin genes resulting in telomere dysfunction, progression of complex chromosomal rearrangements, and multinuclearity [36, 37]. The impact of HHV-6A/B infection on telomere biology is currently unknown. Considering that telomeres are the preferred sites for HHV-6A/B integration, it is important to understand the dynamic processes occurring during the early phases of infection to gain insights into the integration mechanisms. In the present study, we analyzed the impact of HHV-6A/B infections on shelterin complex homeostasis and show for the first time that the shelterin components are recruited to the viral DNA during infection and play a critical role in HHV-6A/B chromosomal integration.

## Materials and methods

### Cell lines and viruses

U2OS cells (American Type culture collection (ATCC), Manassas, VA, USA) were cultured in Dulbecco's modified Eagle's medium (DMEM, Corning Cellgro, Manassas, VA, USA) supplemented with 10% Nu serum (Corning Cellgro), non-essential amino acids (Corning Cellgro), HEPES, sodium pyruvate (Multicell Wisent Inc., St-Bruno, Québec, Canada) and plasmocin 5 μg/ml (InvivoGen, San Diego, CA, USA). Molt-3 (ATCC, CRL-1552), HSB-2 (ATCC, CCL-120.1), Sup-T1 (ATCC, CRL-1942), all human T lymphoblastic cell lines, were cultured in RPMI-1640 (Corning Cellgro) supplemented with 10% fetal bovine serum (FBS), HEPES and plasmocin 5 μg/ml. J-Jhan (RRID:CVCL_1H08) cells infected with wild type (WT) HHV-6A-BAC, HHV-6A mutants lacking telomeric repeats (TMR) termed ΔTMR and ΔimpTMR [6] or HHV-6A BAC WT#2 (containing a red fluorescent protein downstream of the U11 gene) were cultured in RPMI-1640 supplemented with 10% FBS. HHV-6B (Z29 strain) and HHV-6A (U1102 strain) were propagated and titered on Molt-3 and HSB-2 cells respectively, as previously described [38].

### Plasmids

IE2 expression vectors (WT and Δ1290–1500) were previously described [39]. pLPC-MYC-hTRF1 (Addgene plasmid #64164) [40], pLPC-NMYC TRF2 (Addgene plasmid # 16066) [41] and pSXneo 135(T2AG3) (Addgene plasmid#12402) [42] were gifts from Titia de Lange and obtained through Addgene. The generation of pLKO human shTRF2 was previously described [43]. The shTRF2 coding sequence was cloned into the Tet-pLKO-puro vector (Addgene #21915). The Tet-pLKO-puro vector was a gift from Dmitri Wiederschain [44].

### Western blots

Cells were resuspended in Laemmli buffer and boiled for 5 minutes. Samples were loaded and electrophoresed through a SDS-polyacrylamide gel. Samples were transferred onto PVDF

membranes and processed for western blot using rabbit anti-TRF2 (Novus Biologicals), mouse anti-tubulin (Sigma-Aldrich) and rabbit anti-p85 antibodies (Abcam). Peroxidase-labeled goat anti-rabbit IgG and peroxidase-labeled goat anti-mouse IgG were used as secondary antibodies. The Bio-Rad Clarity ECL reagent was used for detection.

## IF-FISH and microscopy

Immunofluorescence (IF) combined with fluorescence *in situ* hybridization (FISH) (IF-FISH) was performed as previously described [45]. U2OS cells were seeded at 5 x $10^4$ cells per well in 6-well plates over coverslips, cultured 24 hours and infected with HHV-6A or HHV-6B at a multiplicity of infection (MOI) of 5 for 4 hours. Cells were then washed with PBS and cultured in media for a set period of time. Cells were fixed with 2% paraformaldehyde. Molt-3 and HSB-2 cells were infected at a MOI of 1 and cultured for a set period of time before being deposited on a 10-well microscope slide, dried and fixed in acetone at -20˚C for 10 minutes. The following primary antibody were used: mouse-α-IE2-Alexa-594 [46], mouse-α-p41 (NIH AIDS Reagent Program), rabbit-α-TRF2 (NB100-56694, Novus Biologicals), and mouse-α-Myc (clone 9E10). Secondary antibodies used were goat-α-rabbit-Alexa-488, goat-α-mouse-Alexa-488 and goat-α-mouse-Alexa-594 (Life Technologies). FISH was performed using a PNA probe specific to the telomeric sequence $(CCCTAA)_3$ (TelC-Cy5, PNA BIO). Slides were observed at 40X and 63X using a spinning disc confocal microscope (Leica DMI6000B) and analyzed with the Volocity 5.4 software.

To compare TRF2 expression in uninfected and HHV-6A-infected cells, cells were dually stained with HHV-6A IE2 protein and TRF2. The relative TRF2 fluorescence in IE2- and IE2 + individual cell was then determined using the ImageJ software.

For colocalization, acquisitions were deconvoluted using the Volocity 5.4 software and Point Spread Function (PSF) respective to the objective and the immersion medium to remove the out-of-focus information from the acquisitions. 3D images were reconstructed using the same software to visualize colocalization. To quantify colocalization of IE2 with telomeres, TRF1, TRF2 or p41, Image J software with JACoP plugin was used. Briefly, after setting up thresholds, total fluorescence of IE2 colocalizing with the fluorescence of telomeres, TRF1, TRF2 or p41was given by Mander's colocalization coefficient (MCC) and reported in percentage were a coefficient of 1 represent 100% of colocalization and 0 equal no colocalization.

## Telomere restriction fragment (TRF) analysis

DNA from uninfected, HHV-6A/B-infected cells and HHV-6A BAC (WT and ΔTMR)-infected cells was isolated using QIAamp DNA blood isolation kits as per the manufacturer's recommendations. Five μg of DNA were digested overnight with RsaI and HinfI followed by electrophoresis through agarose gel and southern blot hybridization. The telomeric DNA probe was obtained following digestion of the pSXneo135(T2AG3) vector with EcoRI and NotI, gel purification of the 820 bp fragment and $^{32}$P-labeling by nick translation. The HHV-6A U94 probe was obtained by digesting the pMalC2-U94A vector [47] with BamHI and HindIII, gel purification of the 1476 bp fragment and $^{32}$P-labeling by nick translation. After hybridization and washes, the membranes were exposed to X-ray films.

## ChIP and dot blot

The experiments were made using the Pierce Magnetic ChIP Kit (Thermo Scientific) according to the manufacturer's instructions with a few modifications. Equal quantities of HSB-2 and Molt-3 cells were used for all samples (4 x $10^6$ cells/sample). Cross-linking lasted 10 minutes at RT. Two μl of diluted MNase (1:10) were added to each sample for MNase digestion.

Sonication was made with a Branson Sonifier 450, with an Output Control set at 1. Each sample was sonicated with five pulses of 20 seconds, each pulse followed by a 20 seconds incubation on ice. After sonication, an aliquot was saved for normalization purpose (input). Before immunoprecipitation, samples were incubated with magnetic beads alone for one hour at 4˚C before discarding the beads. The immunoprecipitation was performed using 4 μg of normal rabbit IgG (negative control), 10 μl of anti-PolII antibodies (positive control) and 4 μg of rabbit anti-TRF2 antibody (NB100-56694, Novus Biologicals) with an overnight incubation at 4˚C. Protein A agarose beads were added for 1h at 4˚C followed by three washes. The DNA was eluted in 50 μl of DNA column elution solution.

Eluted DNA was analyzed by quantitative PCR (qPCR) for GAPDH promoter sequences using reagents and conditions provided by the manufacturers (Pierce Magnetic ChIP Kit, Thermo Scientific) or analyzed by dot blot hybridization using a telomeric probe or HHV-6A probe (DR6). The input was analyzed using an Alu probe. For dot blot hybridization, DNA was first denatured for 10 minutes at room temperature in 0.25 N NaOH and 0.5 M NaCl. Samples were then serially diluted in 0.1 X SSC and 0.125 N NaOH, on ice, loaded onto nylon membrane, neutralized in 0.5 M NaCl and 0.5 M Tris-HCl pH 7.5 and crosslinked using UV irradiation. Membranes were pre-incubated in Perfecthyb Plus hybridization buffer (Sigma-Aldrich) for 2h at 68˚C before addition of 1 x $10^6$ CPM/ml of $^{32}$P-labeled probes. Hybridization was carried out for 16h at 68˚C. Membrane was washed twice with 2X SSC-1% SDS, twice with 1X SSC-1% SDS and once with 0.5X SSC-1% SDS at 68˚C, for 15 minutes each. Membrane was then exposed to X-ray films at -80˚C. Hybridization signals were measured by densitometry.

### Cloning and purification of MBP-TRF2

The TRF2 coding sequence was excised from pLPC-NMYC TRF2 vector using with BamHI and XhoI and cloned in frame with the Maltose-Binding Protein (MBP) coding sequence of the pMAL-C2 vector (New England Biolabs) using BamHI and SalI enzymes. MBP and MBP-TRF2 proteins were expressed in BL21 DE3 RIL bacteria and purified by affinity chromatography, as described [47].

### Electrophoretic mobility shift assay (EMSA)

EMSA was performed essentially as described [47]. In brief, recombinant proteins (MBP and MBP-TRF2) were incubated with 1 pmole of double-stranded (ds) non-telomeric or telomeric labeled probes in 20 μl of the following reaction buffer: 20 mM Hepes-KOH pH 7.9, 150 mM KCl, 1 mM $MgCl_2$, 0.1 mM EDTA, 0.5 mM DTT, 5% glycerol and 0.1 mg/ml BSA. For competition experiments, 10–1000 fold excess unlabeled double-stranded (ds) non-telo or telomeric probes were included in the reaction buffer. After a 30 minutes incubation at room temperature, 2 μl of loading dye were added and the samples were electrophoresed through a non-denaturing 5% acrylamide:bis (29:1) gel. After migration, the gels were dried and exposed to X-ray films at -80˚C.

### Detection of TRF2 binding to HHV-6A telomeric sequence

The wells of a 96-well ELISA plate were coated with 0.6 pmoles of MBP or MBP-TRF2 proteins by overnight incubation at 4˚C in pH 9.0 carbonate buffer. After rinsing, 1% BSA was added to block non-specific sites. Twenty-five nanograms of HaeIII-digested digoxigenin-labeled HHV-6A DNA (HaeIII cuts the viral genome 289 times) (approximately 0.16 pmoles) in EMSA reaction buffer were added. For competition experiments, 2.5 or 5.0 pmoles of non-telomeric or telomeric dsDNA were added 15 minutes prior to the addition of HHV-6A DNA.

The plate was incubated for 2h at room temperature (RT). After 3 washes with TBS-0.1% Tween-20 (TBS-T), peroxidase-labeled mouse anti-DIG antibodies were added to each well for 1h at RT. After 3 additional TBS-T washes, TMB substrate was added and the reaction allowed to develop for 15 minutes before addition of 50 μl of 2N sulfuric acid. Absorbance was measured at 450 nm.

### HHV-6A/B integration assay

HHV-6A/B integration frequency in U2OS cells was determined by droplet digital PCR (ddPCR) as previously described [18]. In brief, cells were infected with either HHV-6A or HHV-6B for 5 h at 37°C and then washed 3× with PBS to remove unadsorbed virions prior to the addition of fresh culture medium. Upon confluence, cells were passaged into the well of a 6-well plate for a few days and further expanded into a 25-cm$^2$ flask for a month until analyzed by ddPCR. ddPCR uses TaqMan chemistry but instead of using a standard curve to estimate copy numbers, it partitions the reaction into thousands of droplets, which are each read as positive or negative for DNA template allowing absolute quantification of DNA copies [48, 49]. HHV-6A/B copy numbers were determined. The HHV-6A/B chromosomal integration frequencies were estimated assuming a single integrated HHV-6/cell and calculated with the following formula: (# of HHV6 copies ÷ (# of RPP30 copies ÷ 2 copies per cell)) × 100. Such a procedure and protocol proved equivalent to estimation of ciHHV-6 frequency using single-cell cloning procedures [18].

### Statistical analysis

TRF2 expression levels were compared using unpaired student t-test with Welch's correction. Binding of MBP-TRF2 to HHV-6A DNA and HHV-6A/B integration frequency were determined using the Mann-Whitney test. Colocalization of IE2 WT and Δ1290–1500 IE2 mutant with telomeres was determined using a t-test. The % of HHV-6A infected cells in shCtrl and shTRF2 +/- Dox was compared using a one-way ANOVA with Tukeys multiple comparisons test. A p value <0.05 was considered significant.

## Results

### Telomeric sequence accumulation during HHV-6A infection

Telomeres protect chromosomes from the loss of genetic information due to end replication problem and the shelterin complex prevents induction of a DNA damage response (DDR). Interestingly, each end of the HHV-6A/B genome also contains telomeric repeats arrays that vary in number between 15 and 180 [9–12] (Fig 1A). During infection, viral replication compartments (VRC) can be visualized by staining cells with antibodies against the DNA polymerase processivity factor p41, encoded by the *U27* gene, that associates with the viral DNA during replication (Fig 1B). The majority (92% ± 7%) of HHV-6A immediate-early 2 (IE2) protein colocalizes with the p41 protein and is used in subsequent experiments as marker of VRC [50]. Using immunofluorescence combined with fluorescent *in situ* hybridization (IF-FISH), we first studied the accumulation of telomeric sequences during active HHV-6A infection. Hybridization of mock-infected HSB-2 cells with a telomeric probe resulted in the detection of many discrete punctate telomeric signals corresponding to terminal chromosomal telomeric repeats (Fig 1B and 1C). In contrast, a mixture of small and large telomeric signals were observed during HHV-6A infection. At late stages of infection when viral genome replication is abundant, very intense telomeric signals were detected. These telomeric signals correspond to replicating virus genomes as they localize with the p41 (Fig 1B) and IE2 protein (Fig

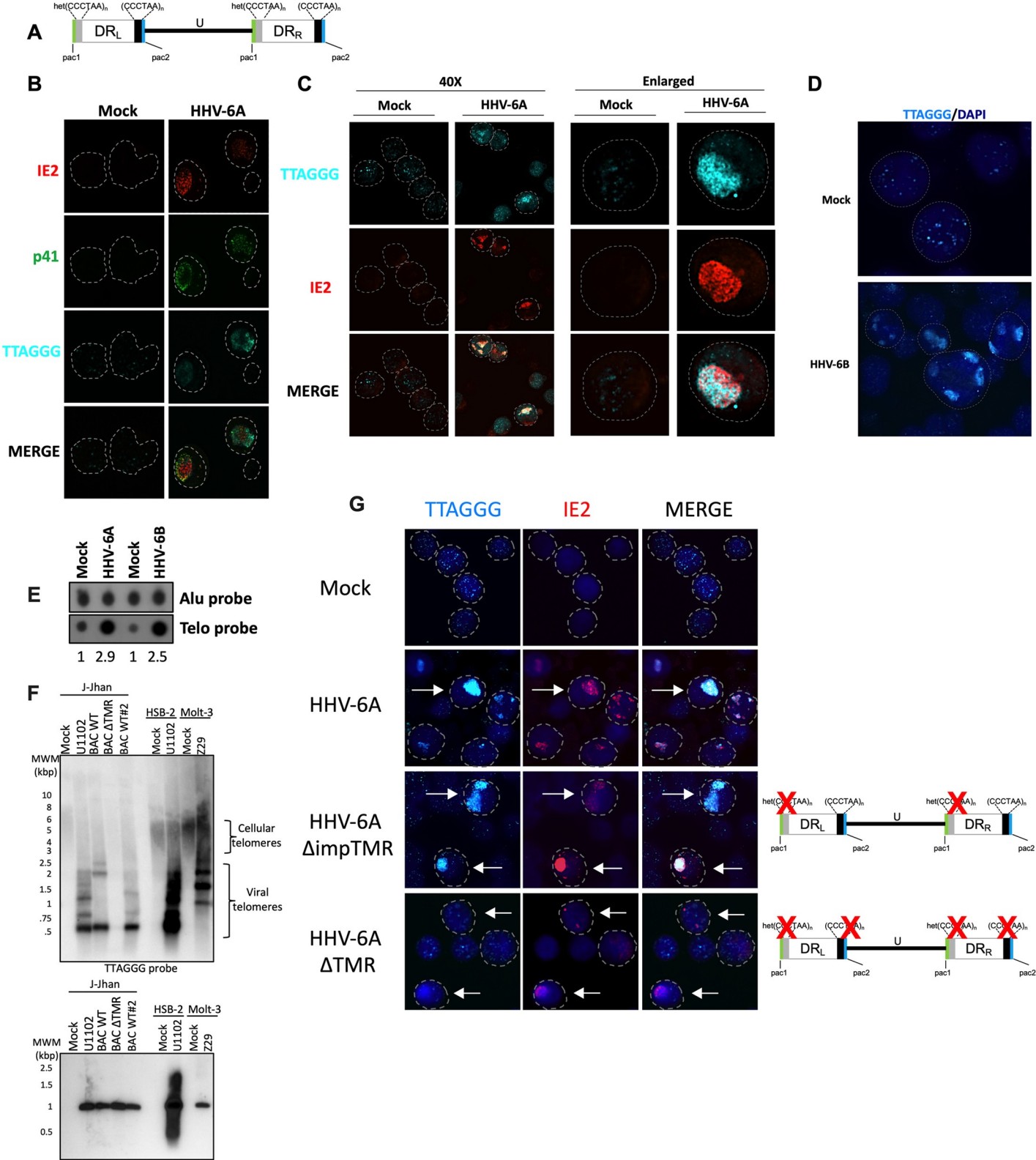

**Fig 1. Accumulation of viral telomeric signals during HHV-6A infection.** A) Schematic representation of the HHV-6A/B genome. The unique (U) region of the HHV-6A/B genome (143–145 kbp) is flanked by two direct repeat sequences (8–10 kbp) referred to as $DR_L$ and $DR_R$. The DRs contain perfect $(TTAGGG)_n$ and imperfect (het $(TTAGGG)_n$) telomeric sequences. The genome is not drawn to scale. B) HSB-2 cells were mock treated or infected with HHV-6A. After 5 days, cells

were processed for IF-FISH to detect HHV-6A IE2 protein (red), p41 (green) and telomeres (cyan) to appreciate HHV-6A VRC. Nuclei are outlined by the circular dashed lines. C) Mock or HHV-6A-infected HSB-2 cells were processed for IF-FISH to detect IE2 and telomeric signals patterns in presence of HHV-6A. Low and high magnifications of cells are presented. Nuclei are outlined by circular dashed lines. D) Molt-3 cells were infected with HHV-6B for 4 days and hybridized with a telomeric probe (cyan) with nuclei stained with DAPI. Nuclei are outlined by circular dashed lines. E) HSB-2 and Molt-3 cells were respectively infected with HHV-6A and HHV-6B. After 5 days, total DNA was isolated and analyzed by dot blot hybridization using Alu and telomeric probes. The relative increase in telomeric signals was determined by densitometry after normalization to uninfected controls. F) HSB-2 and Molt-3 cells were respectively infected with HHV-6A and HHV-6B. J-Jhan cells were infected with HHV-6A U1102, WT HHV-6A BACs or HHV-6A BAC ΔTMR. After 4 days, DNA was extracted, digested with HinfI and RsaI and processed for southern blot hybridization using $^{32}$P-labeled telomeric and U94 probes. G) J-Jhan cells were infected with HHV-6A U1102, WT recombinant HHV-6A BACs, HHV-6A mutant lacking the imperfect telomeric repeats (ΔimpTMR) or HHV-6A mutant lacking all telomeric repeats (ΔTMR). After 5 days of infection, cells were processed for IF-FISH to detect HHV-6A IE2 protein and telomeres (cyan). Some infected cells expressing IE2 are identified by arrows. Nuclei were stained with DAPI and outlined by dashed lines.

1C). Similar accumulation of telomere signal resembling VRC were observed in HHV-6B-infected cells (Fig 1D). Considering that our IE2 antibody is specific for HHV-6A, staining for HHV-6B IE2 was not possible [46, 51]. Next, the increase in telomeric signals observed by IF-FISH was confirmed by dot blot hybridization. DNA from uninfected and HHV-6A/B infected cells was hybridized with a telomeric probe to estimate overall telomeric repeats. Hybridization with an Alu probe was used for normalization. HHV-6A/B infection resulted in a 2.5x to 2.9x increase in the total number of telomeric signals relative to uninfected cells (Fig 1E). To determine the origin (cellular or viral) of the increased telomeric signals, DNA from uninfected and HHV-6A/B infected cells was analyzed by terminal restriction fragment (TRF) analysis and southern blot hybridization with a telomeric probe. As shown in Fig 1F, HSB-2 and Molt-3 uninfected cells displayed telomeric signal with lengths ≥4kbp. In addition to the cellular telomeric signal observed, HHV-6A/B-infected cells displayed abundant signals that were smaller in size (<2kbp) and much stronger in intensity, likely representing viral telomeric repeats. To confirm the viral origin of these telomeric repeats, J-Jhan cells were infected with HHV-6A U1102 (control), WT HHV-6A-BACs or HHV-6A ΔTMR BAC that lacks the viral telomeric sequences [6]. In J-Jhan cells, low molecular weight telomeric signals were detected in HHV-6A and WT HHV-6A BACs but were absent in cells infected with HHV-6A ΔTMR BAC confirming the viral origin of telomeric signal. Infection was confirmed by hybridizing the membrane with a probe corresponding to the HHV-6A *U94* gene, expected to hybridize to a 964 bp viral DNA fragment.

The accumulation of telomeric sequences was also confirmed by IF-FISH using J-Jhan cells infected with HHV-6A recombinant BACs. As with HSB-2 cells, mock-infected J-Jhan showed typical telomeric staining (Fig 1G, first row). J-Jhan cells productively infected with HHV-6A demonstrated large telomeric signals that colocalized with the viral IE2 protein (second row). To demonstrate that the increased telomeric signals observed originates from viral DNA, we made use of HHV-6A mutants lacking either only the imperfect telomeric repeats (ΔimpTMR) or all telomeric repeats (ΔTMR) [6]. Infection with the ΔimpTMR mutant still resulted in a strong and patchy telomeric signals (Fig 1G, third row). In contrast, telomeric hybridization signals in ΔTMR-infected J-Jhan cells were similar to those observed in uninfected J-Jhan cells (Fig 1G last row), confirming that telomeric sequences within the viral genome were responsible for the increased telomeric signals observed.

## Binding of TRF2 to viral telomeric sequences

Telomeres are protected by the shelterin complex of which TRF1 and TRF2 bind directly to TTAGGG repeats; however, it remains unknown if shelterin proteins bind to viral telomeric sequences in the context of the HHV-6A/B genomes. To study TRF2 binding to viral TMRs, a recombinant MBP-TRF2 protein was generated. To validate that MBP-TRF2 was functional and capable of binding telomeric DNA, we performed EMSA. MBP-TRF2 efficiently bound

dsDNA with telomeric sequences causing a mobility shift (Fig 2A). MBP alone did not bind the telomeric probe. The specificity of MBP-TRF2 binding was confirmed by competition with excess unlabeled telomeric and non-telomeric oligonucleotides. Excess (100–1000 fold) of unlabeled telomeric oligonucleotides efficiently competed with labeled telomeric probes. No such competition was observed with excess non-telomeric oligonucleotides. Lastly, no binding of MBP or MBP-TRF2 was observed using non-telomeric labeled probes (Fig 2B).

After validation of the specific binding to telomere sequences of the recombinant MBP-TRF2 protein, we next determined if MBP-TRF2 could bind to HHV-6A TMR DNA. To study this, DIG-labeled HHV-6A-BAC DNA was digested with the HaeIII enzyme that cuts on both sides of the viral TMR and more than 250 times in the viral genome. MBP and MBP-TRF2 coated wells were incubated with the mixtures of DNA fragments (25 ng) and DNA binding was measured using anti-DIG antibodies. MBP did not bind viral DNA, in contrast to MBP-TRF2 that efficiently bound viral DNA (Fig 2C). Specificity of MBP-TRF2 binding to viral TMR was confirmed through competition with unlabeled ds oligonucleotides containing telomeric motifs (Telo comp) but not by ds oligonucleotides with non-telomeric motifs (non Telo comp). Our *in vitro* binding assay revealed that the recombinant MBP-TRF2 efficiently binds to viral DNA at TMRs.

To provide additional support that TRF2 binding occurs during infection, we performed TRF2 ChIP in HHV-6A and HHV-6B productively-infected cells. Uninfected cells were used as negative controls. To discriminate between telomeres of cellular and viral origin, the TRF2 immunoprecipitated DNA was hybridized with the DR6 probe, corresponding to regions adjacent (1.5kbp) to the TMR in the virus genome (refer to Fig 3A). As positive control, binding of PolII to the GAPDH promoter was analyzed by qPCR. As shown in Fig 3B, GAPDH promoter sequences were highly enriched following anti-PolII precipitation. Next, DNA bound to TRF2 was immunoprecipitated (IP) using anti-TRF2 antibodies and the corresponding DNA analyzed by dot blot hybridization. Hybridization with a telomeric probe indicates that TRF2 efficiently bound telomeres in both uninfected and HHV-6A- and HHV-6B-infected cells (Fig 3C–3F). Compared to uninfected cells, a stronger telomeric signal was observed in infected cells. TRF2 also precipitated DNA that hybridized preferentially (10-fold) with the DR6 probe in HHV-6A-infected cells (Fig 3C and 3D). As negative control, DNA was immunoprecipitated with an irrelevant mouse anti-IgG. Similar results were obtained for HHV-6B (Fig 3E and 3F). In summary, these assays provide evidence that TRF2 binds the telomeric motifs present in HHV-6A/B DNA during productive HHV-6A/B infections.

## HHV-6A and shelterin during infection of U2OS cells

Considering that HHV-6A/B integration invariably occurs in telomeres of infected cells, we surmise that shelterin proteins likely play a role in viral integration. As most T cells are lytically infected and subsequently killed, HHV-6A/B integration does not occur frequently in these cells. We therefore studied the fate of shelterin proteins in HHV-6A/B-infected U2OS cells that are semi-permissive to infection and used to study viral integration [18]. After 24h, 48h and 72h post-infection, individual cells were analyzed for TRF2 expression. Infected cells were distinguished from uninfected bystander cells using the HHV-6A specific anti-IE2 antibody (Fig 4A). Depending on the time of infection, between 38% (24 h post infection) and up to 60% (72h post-infection) of IE2 colocalized with TRF2. TRF2 fluorescence was quantified using ImageJ software. The results obtained indicate that starting at 24h post-HHV-6A infection, TRF2 expression gradually increases as infection progresses. TRF2 is expressed at significantly higher levels in infected cells relative to bystanders or uninfected cells ($p \leq 0.02$) (Fig 4B). No significant difference in TRF2 expression was detected between bystander and mock-

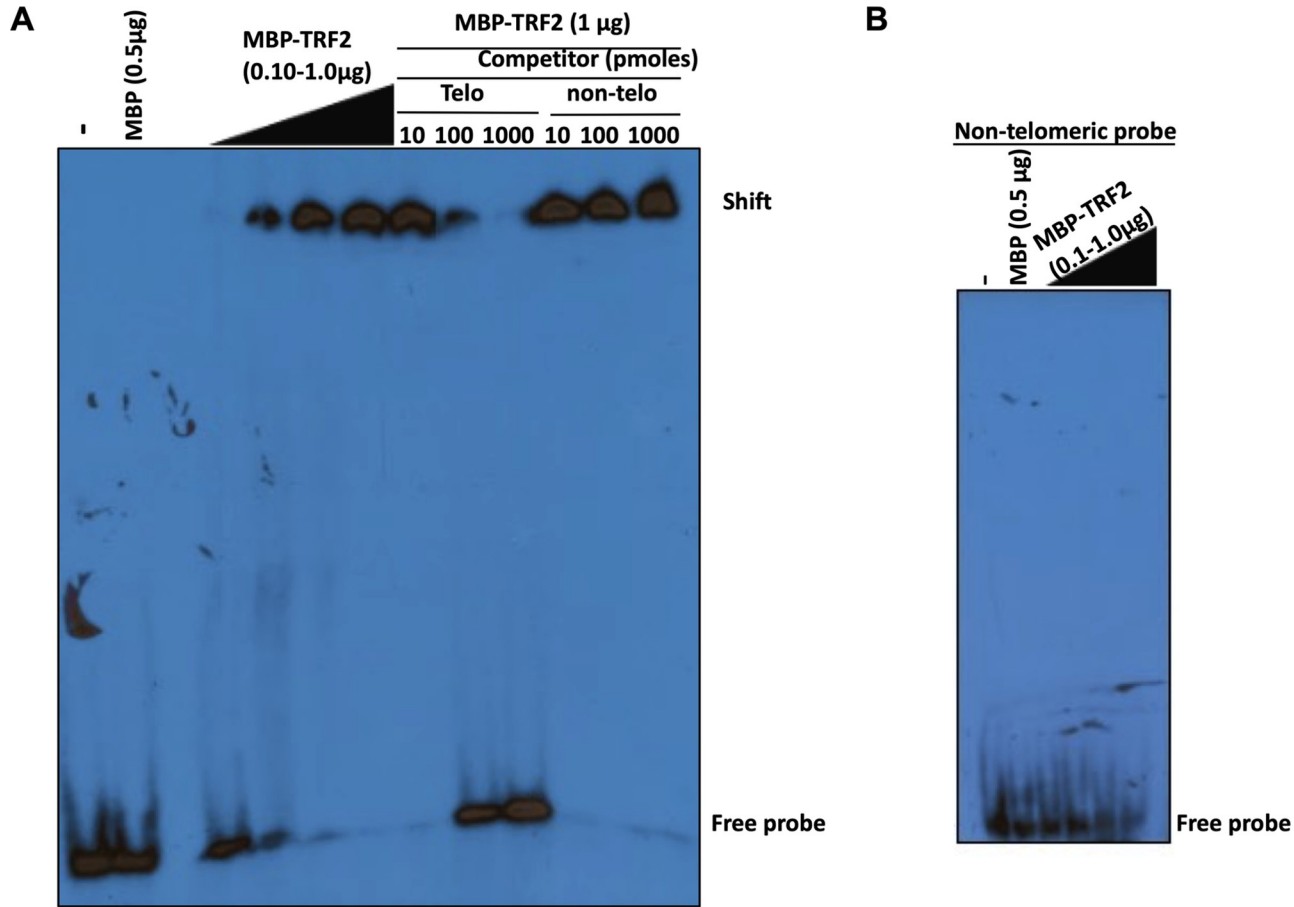

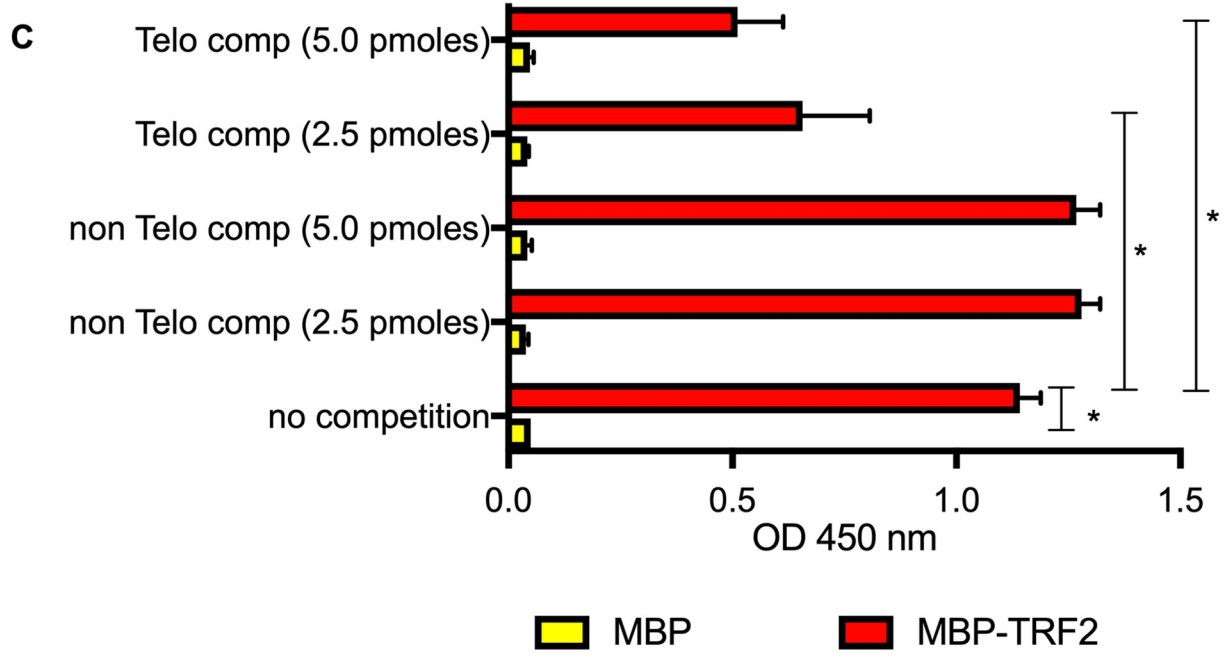

**Fig 2. Binding of TRF2 to HHV-6A viral DNA.** A) Recombinant MBP or MBP-TRF2 were incubated with $^{32}$P-labeled telomeric dsDNA and binding was assessed by EMSA. Excess of unlabeled telomeric and non-telomeric dsDNA were added as competitors. Samples were migrated on non-denaturing acrylamide gel, dried and exposed to X-ray films. B) Recombinant MBP or MBP-TRF2 were incubated with $^{32}$P-labeled non-telomeric dsDNA and binding was assessed by EMSA. C) Recombinant MBP and MBP-TRF2 were coated to the wells of a 96 well-plate and incubated with HaeIII digested DIG-labeled HHV-6A DNA (25 ng/condition) in the presence or absence of competitors. After washing, bound DNA was quantified by adding peroxidase-labeled anti-DIG antibodies and substrate. Results are expressed as mean absorbance +SD of triplicate values. Experiment is representative of two additional experiments. * P<0.001.

treated cells. In summary, these results indicate that expression of TRF2 increases during HHV-6A-infection of U2OS cells.

Colocalization of TRF2 with viral DNA during infection of U2OS cells was studied next. Although permissive to entry and expression of viral genes, viral DNA replication is observed only in a minority of HHV-6A/B infected U2OS cells. As shown in Fig 5A (middle row), some cells display large patchy IE2 staining similar to what is observed during productive infection of T cells (Fig 1B and 1C) and likely representing VRC. Such IE2 patches overlapped with diffuse TRF2 and telomeric signals. Uninfected cells (top row) only showed punctate and sharp TRF2 and telomeric signals. In the majority of cells, (bottom row), the presence of diffuse telomeric signals and "patchy" IE2 was not observed. In these cells, IE2 is present as small punctate structures. On average, 47%±25% of punctate IE2 colocalized with TRF2 and telomeres.

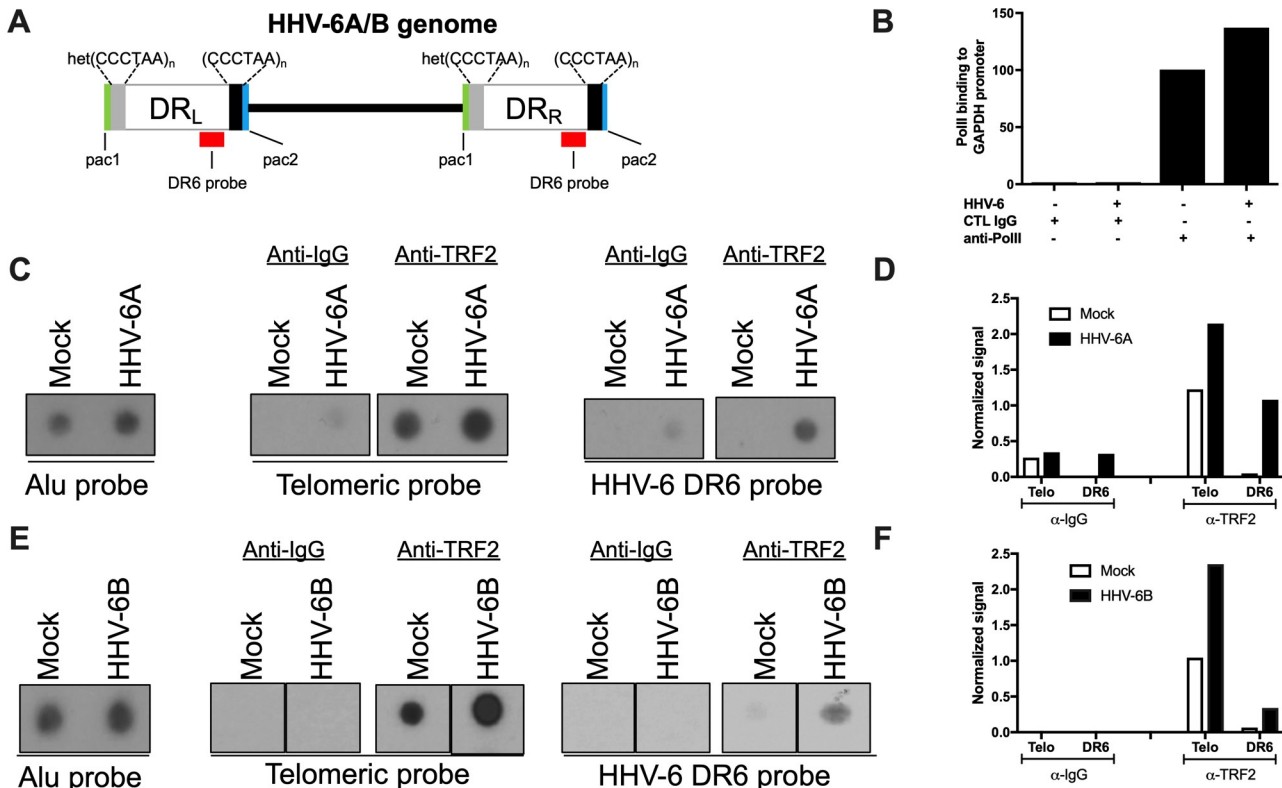

**Fig 3. Binding of TRF2 to viral DNA during HHV-6A/B infection.** A) Schematic representation of the HHV-6A/B genome. The DR6 probe used for hybridization is shown in red. Uninfected and HHV-6A-infected HSB-2 cells (B-D) or uninfected and HHV-6B-infectd Molt-3 cells (E-F) were analyzed for TRF2 binding to viral DNA using ChIP. The input was hybridized with Alu probe to assess quantity of starting material. Anti-IgG (negative control), anti-PolII (positive control) or TRF2 antibodies were used for immunoprecipitation. B) QPCR detection of GAPDH DNA following ChIP. Results are expressed as fold increase over control IgG. C and E) Eluted DNA was hybridized with $^{32}$P-labeled Alu, telomeric (TTAGGG)$_3$ or HHV-6A (DR6) probes. After hybridization the membranes were washed and exposed to X-ray films. D and F) Densitometric analysis of relative binding of TRF2 to telomeric and viral DNA. Results of one experiment representative of three are presented and are expressed as signal after normalization to input.

**A**

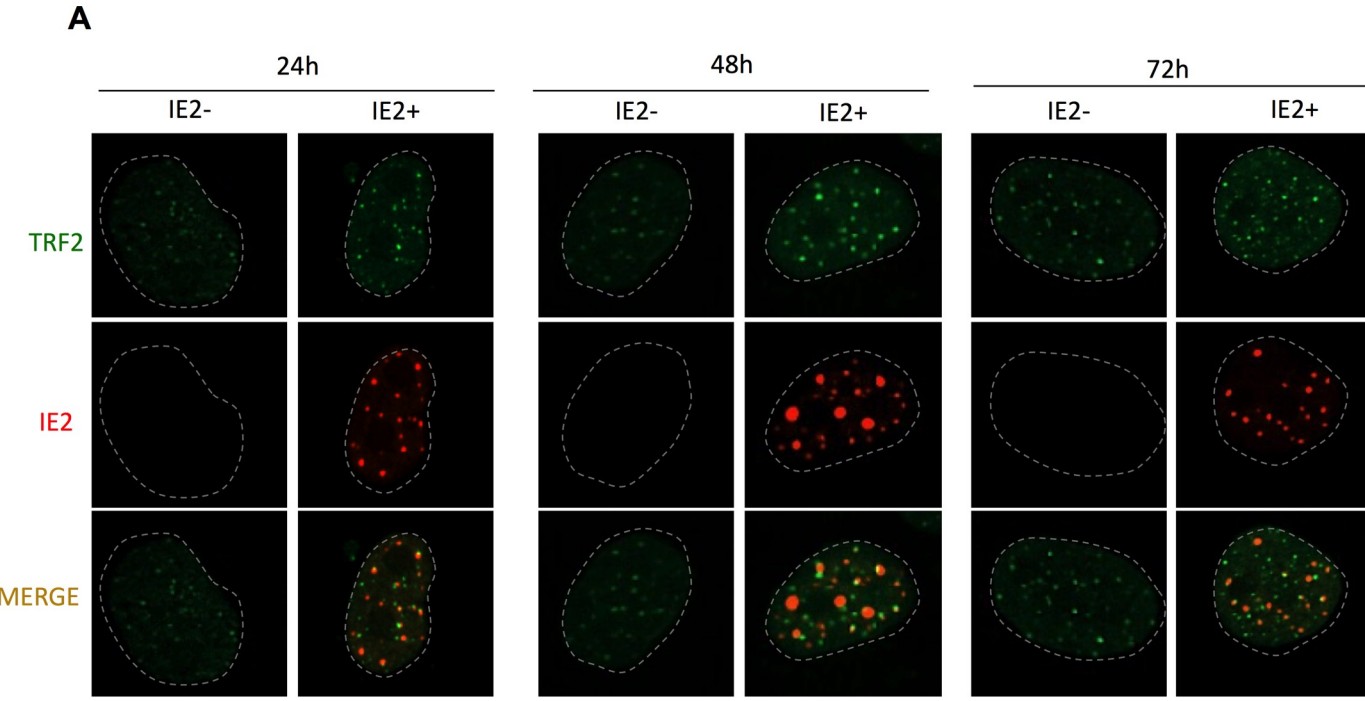

**B**

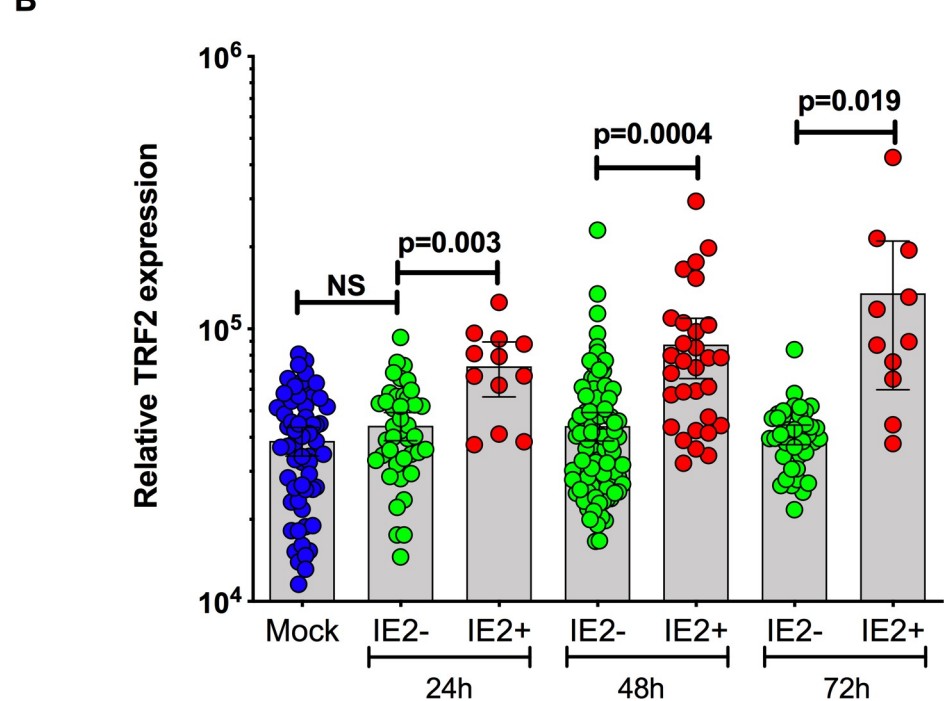

**Fig 4. TRF2 expression in HHV-6A-infected U2OS cells.** U2OS cells were infected with HHV-6A and analyzed for TRF2 and IE2 expression at 24h, 48h and 72h post-infection by dual color immunofluorescence. A) Representative immunofluorescence of TRF2 and IE2 expression in bystander and IE2 expressing cells at 24, 48h and 72h post infection. B) Mean relative TRF2 expression ± SD in uninfected (blue), IE2- (green-uninfected bystander) or IE2+ (red-infected) cells at 24h, 48h and 72h post infection. Each symbol represents the relative TRF2 expression from a single nucleus.

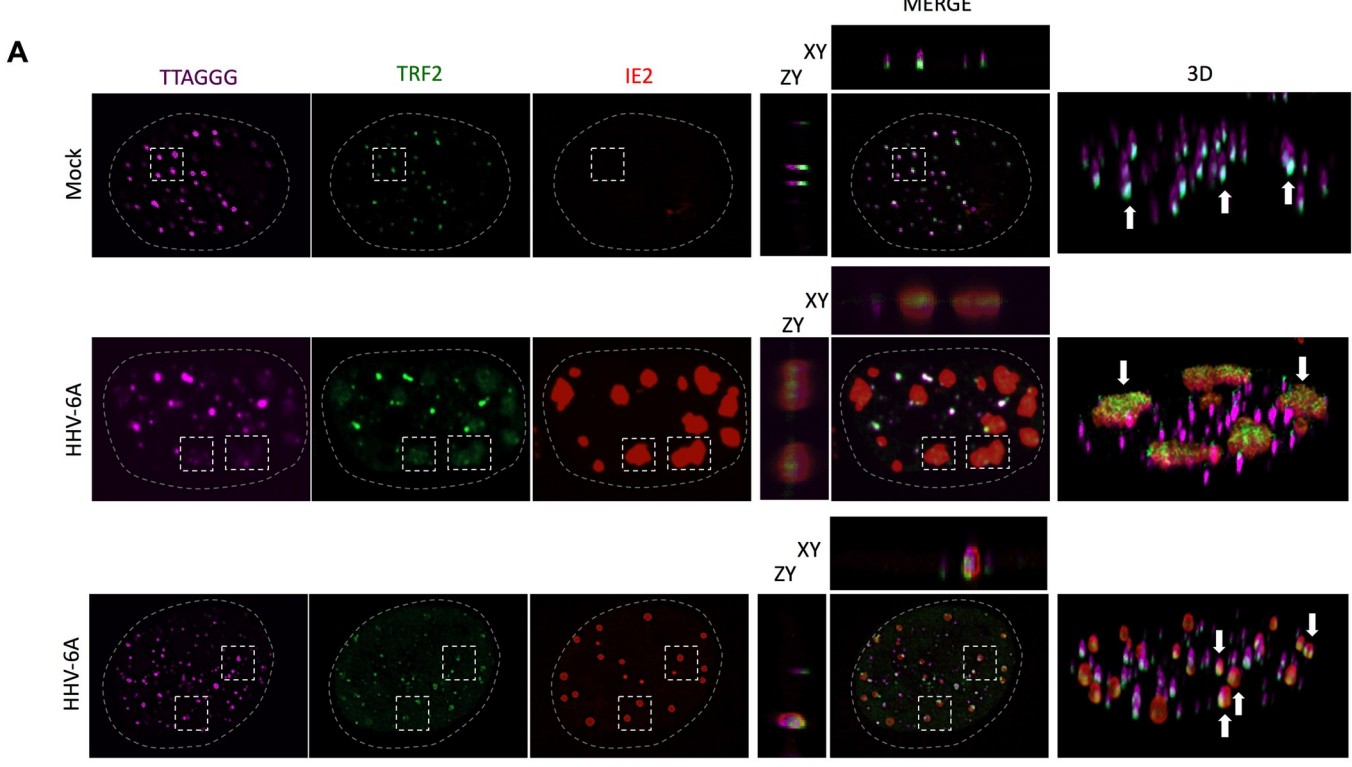

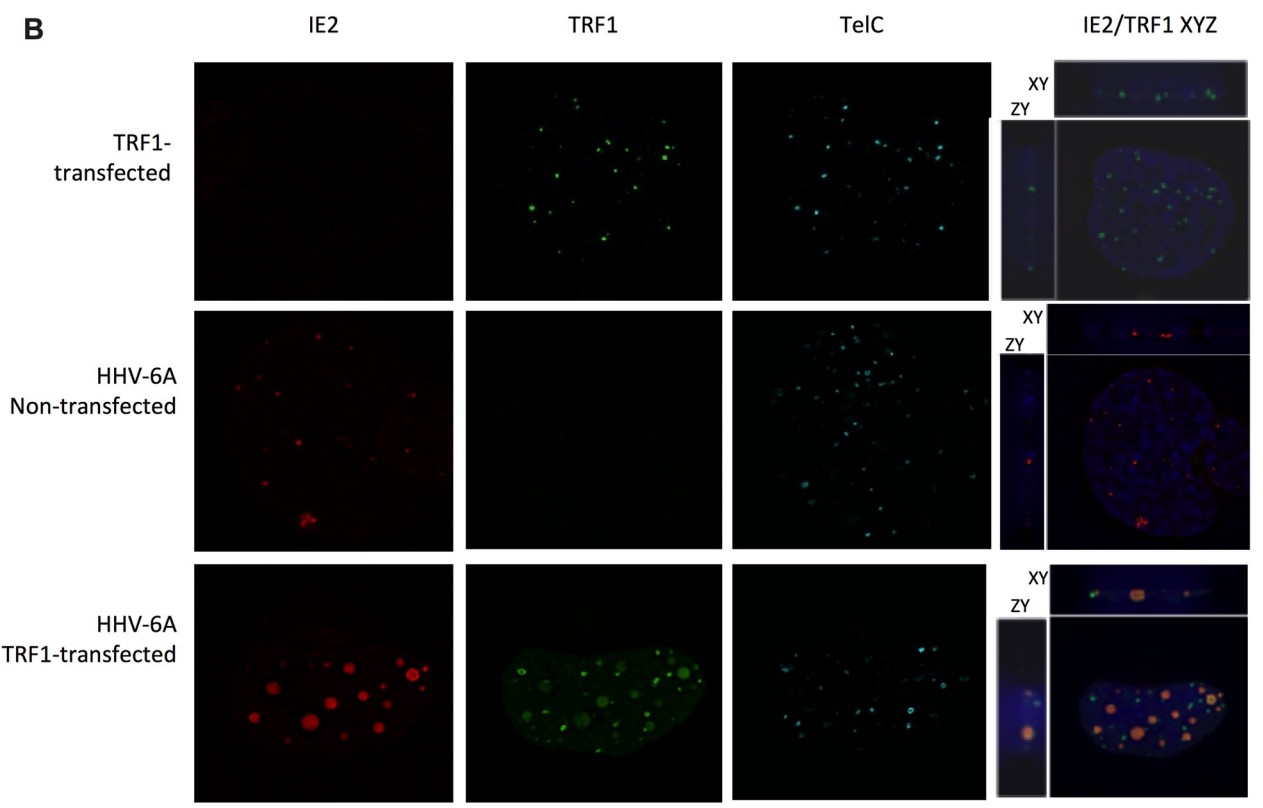

**Fig 5. Colocalization of shelterin complex proteins and HHV-6A IE2 protein at VRC and cellular telomeres.** A) U2OS cells were mock-infected or infected with HHV-6A for 48h after which cells were processed for IF-FISH. Telomeres were labeled in magenta, TRF2 in green and IE2 in red. The panels in the middle row show images of cells with IE2 patches overlapping with large, diffuse TRF2 and telomeric staining (rectangles). The panels in the third row represent infected cells with punctate IE2 pattern colocalizing with TRF2 and telomeres (dashed squares). The colocalization of IE2, TRF2 and telomeres are shown in both 2D and 3D images. B) Uninfected and HHV-6A-infected U2OS cells were transfected with an empty vector, a myc-tagged-TRF1 expression vector. Forty-eight hours later cells were processed for IF-FISH. TRF1 was labeled in green and IE2 in red. Nuclei were stained with DAPI. Images on the far right show 2D colocalization of TRF1 with IE2.

We next investigated whether TRF1, another shelterin protein binding to TTAGGG repeats, colocalizes with HHV-6A DNA during infection. U2OS were transfected with a myc-tagged-TRF1 expression vector, infected with HHV-6A and analyzed by IF-FISH. We used an expression vector to express a tagged version of TRF1 since available antibodies against TRF1 were not specific. As shown in Fig 5B, in addition to its typical punctate pattern, TRF1 also was distributed in diffuse patches that colocalized with IE2/VRC (78% ±20%)

Results from Figs 4 and 5 indicated that a significant proportion of IE2 colocalizes with TRF1, TRF2 and telomeres during infection. We next determined whether ectopically-expressed IE2 (Fig 6A) would also localize with telomeres in the absence of other viral proteins and viral DNA. U2OS were transfected with an empty vector or an IE2 expression vector and cells were analyzed by IF-FISH 48h later. In the absence of viral DNA, IE2 always distributes itself with a punctate nuclear distribution. In average, 58%±30% of IE2 foci colocalized with cellular telomeres (Fig 6B and 6C). Considering that IE2 possesses a DNA-binding domain (DBD), its telomeric localization was studied next. The IE2 mutant (Δ1290–1500) lacking the DBD localized with telomeres with equal efficiency (57%±22%) as WT IE2, indicating that the DBD is dispensable for IE2 localization with telomeres (Fig 6B and 6C). There were no statistically significant differences (p>0.05) in the telomere colocalization frequencies between IE2 expressed during infection and ectopically expressed IE2. Lastly and as further control, we analyzed the nuclear distribution and colocalization of ectopically expressed p41 viral protein with TRF2. As shown in Fig 6D, ectopically-expressed p41 exhibited mostly a perinuclear distribution that did not colocalize with cellular telomeres.

## TRF2 is essential for efficient localization of IE2 at cellular telomeres

Considering that HHV-6A IE2 colocalizes with TRF2 at cellular telomeres and with TRF2 at VRC, we hypothesized that TRF2 might influence IE2 localization. We generated a U2OS cell line carrying a doxycycline (Dox)-inducible shRNA targeting TRF2 mRNA. Incubation of cells with Dox for 7 days resulted in TRF2 knockdown (Fig 7A and 7B). In both control and TRF2 knockdown cells, the IE2 nuclear distribution, whether as punctate foci or patches, was similar (Fig 7B). The percentage of HHV-6A-infected cells was also equivalent in the presence or absence of TRF2 (Fig 7C). However, when the localization of IE2 at cellular telomeres was estimated, we observed a significant reduction of IE2 at cellular telomeres in the absence of TRF2 (Fig 7D). In shCtrl cells (+Dox), 65±23% of IE2 foci were found localizing with telomeres while in shTRF2 (+Dox) treated cells, 13±6% of IE2 foci were found with telomeres (p<0.0001). Confocal IF-FISH revealed the lack of localization of IE2 with telomeres in the absence of TRF2 (Fig 7E).

## Importance of TRF2 for efficient HHV-6A/B chromosomal integration

Considering that TRF2 colocalizes and interacts with viral DNA during HHV-6A/B infections and that IE2 localization at cellular telomeres is influenced by TRF2, we next determined whether TRF2 knockdown would affect HHV-6A/B chromosomal integration. U2OS were transduced with lentiviral vectors constitutively expressing a scrambled shRNA (shCtrl) or

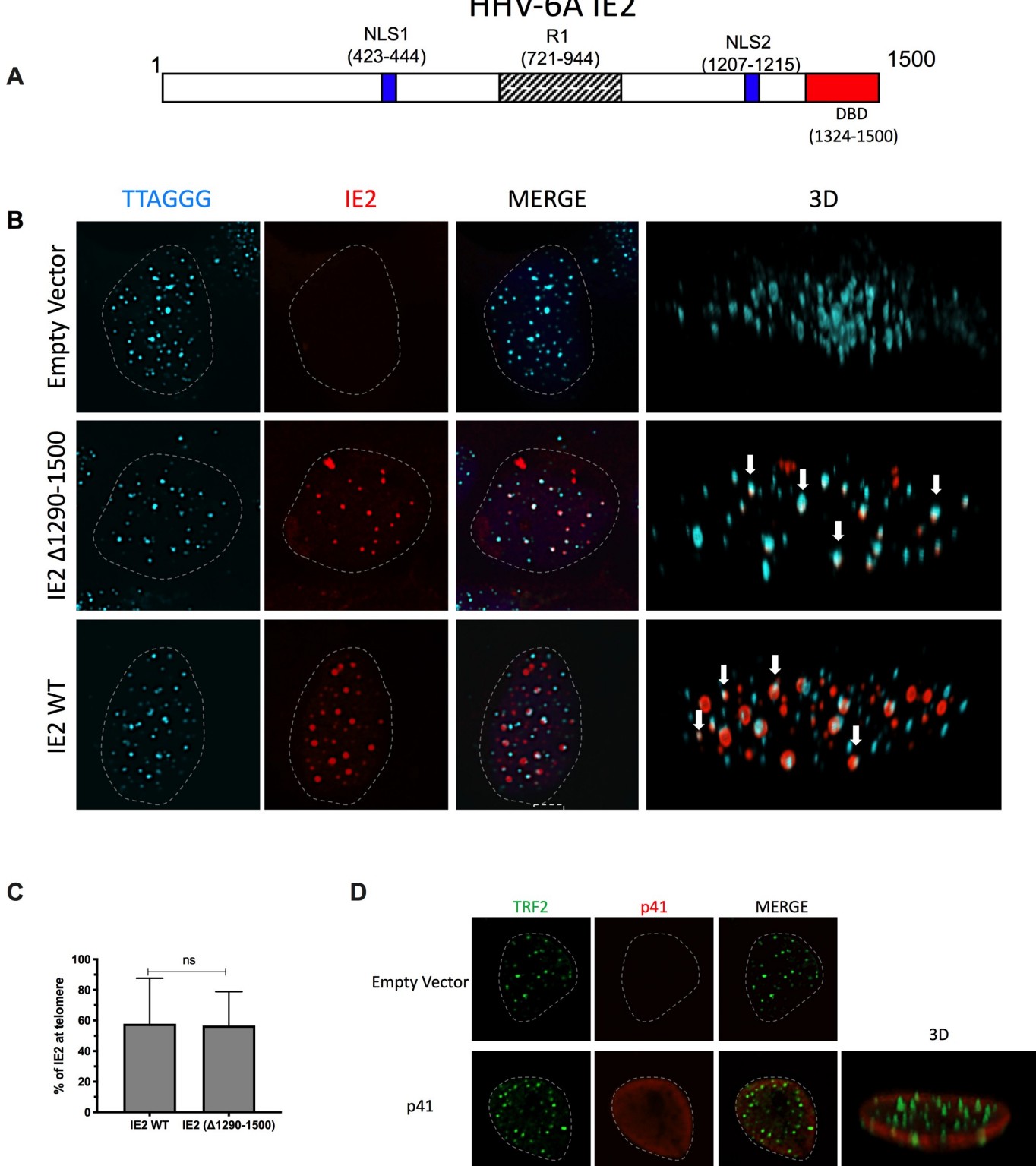

**Fig 6.** A) A stick diagram of the IE2 protein with various domains identified is presented. B) Colocalization of HHV-6A IE2 protein with telomeres in the absence of viral DNA. U2OS cells were transfected with an empty vector, with IE2 expression vector or with IE2 Δ1290–1500 expression vector. Forty-eight hours later cells were processed for dual color immunofluorescence. Telomeres were labeled in cyan and IE2 in red. Nuclei are outlined by dashed lines. Examples of IE2 colocalizing with telomeres are presented in a 3D view (white arrows). C) The graph represents the mean ± SD % of WT IE2 and Δ1290–1500

IE2 localizing with telomeres. D) Lack of colocalization between HHV-6A p41 and telomeres in uninfected cells. U2OS cells were transfected with an empty vector or with a p41 expression vector. Forty-eight hours later cells were processed for dual color immunofluorescence. TRF2 was labeled green, p41 in red and nuclei outlined by a dashed line.

shTRF2. After a week of puromycin selection, TRF2 expression and efficiency of knockdown was monitored by western blot analysis. Compared to the shCtrl, TRF2 expression was significantly reduced by the shTRF2 (Fig 8A). Control and TRF2 knockdown cells were then infected with HHV-6A or HHV-6B and infections allowed to proceed for a month, after which cells were analyzed by ddPCR to assess relative HHV-6A/B integration frequency, as previously described [18]. Integration frequency of HHV-6A and HHV-6B was reduced by more than 75% (p<0.05) in TRF2 knockdown cells compared to the shCtrl control (Fig 8B), suggesting that TRF2's presence is required for efficient HHV-6A/B chromosomal integration.

## Discussion

Telomeres serve to protect chromosomes from the loss of genetic information. The ends of each chromosome contain several hundreds, even thousands, of tandemly repeated TTAGGG hexamers. Each time a cell divides approximately 150 nucleotides are lost due to the end replication problem [52]. When telomeres get short, these are extended either by the telomerase enzyme complex [53] or alternative lengthening mechanisms [54]. On the other hand, when telomeres get excessively long, proteins such as TZAP, can trim the excess telomeres [45]. Mechanisms sensing the length of telomeres are therefore present in cells to control telomere length. In the present study, we report that during HHV-6A/B infection, the number of TTAGGG repeats per cell increases 2.5 to 2.9 fold. The increase in telomeric sequences originates from replication of the viral genomes that contain between 15 and 180 TTAGGG repeats at each viral extremity [9–12]. An HHV-6A mutant lacking these telomeric sequences did not show this phenotype.

Chromosome telomeres are consistently bound by shelterin proteins that serve to protect the linear chromosome end from inappropriate repair through DDR [23]. Three shelterin proteins, TRF1, TRF2 and POT1 recognize and bind specifically to the TTAGGG motif [26, 32, 33]. Shelterin protein binding to DNA of viruses other than HHV-6A/B has been reported previously. Binding of TRF2, TRF1 and Rap1 to EBV oriP, that contains three TTAGGGTTA motifs, was reported to modulate EBV DNA replication. TRF2 also interacts with EBNA1, an EBV protein essential for episomal maintenance and replication [55]. While TRF2 and Rap1 promote the replication at oriP, TRF1 inhibits it [55–57]. TRF2, together with Kaposi sarcoma-associated herpesvirus (KSHV) LANA protein bind to the latent origin of replication. Such region does not contain the TTAGGG motif and binding to this region of the viral DNA likely involves a yet to be identified protein [57]. Unlike EBV, the expression of a dominant negative TRF2 does not affect KSHV DNA replication. Whether the presence of telomeric repeats within the HHV-6A/B viral genome would initiate the binding of shelterin proteins was unknown. Our results indicate that during infection, TRF1 and TRF2 localize at viral replication compartments. Using ChIP, we could demonstrate that TRF2 physically associates with HHV-6A/B viral DNA during infection. Furthermore, using recombinant TRF2 and HHV-6A BAC viral DNA, we could show that TRF2 binds directly to viral DNA TMR in the absence of other viral or cellular proteins.

Joining TRF2 at viral replication compartments are the p41 and IE2 viral proteins. p41 is expected to localize with VRC being a DNA polymerase accessory factor. In contrast, the functions of IE2 are only partially known. IE2 is a large nuclear protein (~1500 amino acids) that behaves as a promiscuous transactivator in gene reporter assays [39, 50]. We have previously

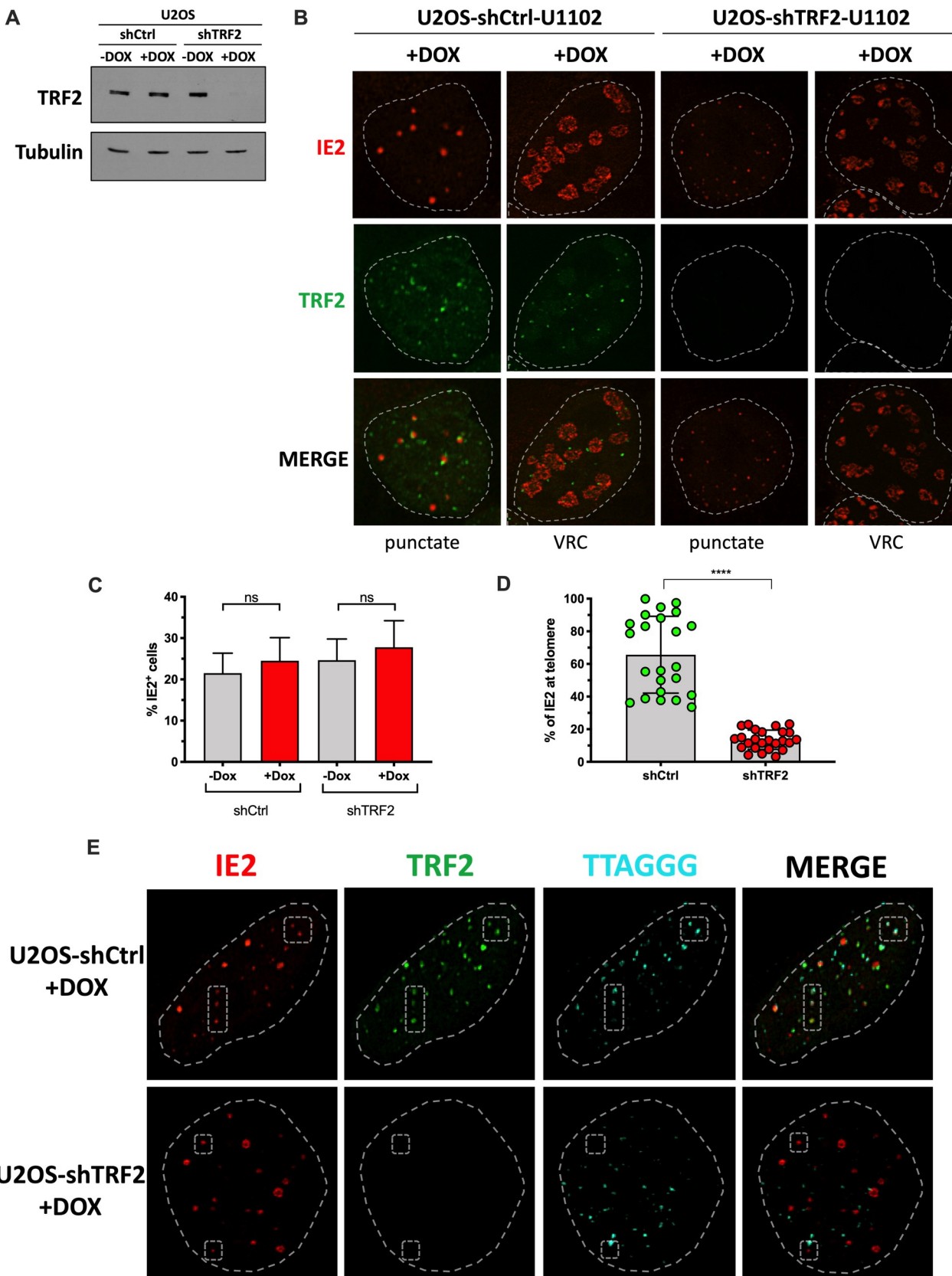

**Fig 7. TRF2 is required for IE2 localization with telomeres.** U2OS cells were transduced with a lentiviral vector coding for a Dox inducible control shRNA (shCtrl) or a shRNA against TRF2 (shTRF2) and selected with puromycin +/- Dox for a week. A) Western blot analysis of TRF2 expression one week post selection. Membranes were also probed with anti-tubulin antibodies to show the input material loaded. B) One week post selection, +Dox cells were infected with HHV-6A for 48h and processed for IFA using anti-TRF2 (green) and anti-IE2 (red). Cells with IE2 in punctate form and cells with large patchy IE2, likely to represent VRC, are shown. Nuclei are outlined by dashed lines. C) The percentage of HHV-6A infected cells (from B) was estimated after counting a minimum of 700 cells and scoring the IE2⁺ ones. Results are expressed as mean %IE2⁺ cells ± SD. D) Mean percentage ± SD of IE2 localizing with telomeres in the presence (shCtrl +Dox) and absence (shTRF2 +Dox) of TRF2. Each dot represents the % of IE2 foci localizing with telomeres in one nucleus. ****p<0.0001. E) IF-FISH confocal images of shCtrl (+Dox) and shTRF2 (+Dox) cells analyzed for TRF2 (green), IE2 (red) and telomeres (cyan). Nuclei are outlined by dashed circles. Examples of IE2 localizing with telomeres (top row) or not found with telomeres (bottom row) are highlighted by the dashed polygons.

reported that truncation of the C-terminus abolishes IE2's transactivating potential [39]. Recently, the crystal structure of the IE2 C-terminus revealed that it contains dimerization, DNA-binding and transcription factor binding domains explaining the importance of this region for IE2's transactivation functions [58]. The IE2 C-terminus core structure resembles those of the gammaherpesvirus factors EBNA1 of EBV and LANA of KSHV [58], involved in binding to viral DNA [59, 60]. Although IE2 localizes at VRC during infection, whether it binds viral DNA *per se* remains to be demonstrated. However, considering that IE2 localizes at VRC in cells infected with a mutant lacking telomeric repeats (HHV-6A ΔTMR), suggests that IE2 recruitment at VRC is independent of viral telomeric DNA sequences. Furthermore, considering that in the absence of viral DNA, IE2 localizes at cellular telomeres, suggest a potential affinity of IE2 for shelterin complex proteins. Deletion of the IE2 DNA binding domain had no impact on IE2 localization with telomeres, further strengthens the hypothesis the IE2 interacts with proteins found near or associated with telomeres. In support for an IE2-shelterin interaction is the observation that TRF2 knockdown greatly reduces the number of IE2 foci localizing with telomeres. Validation of a physical interaction between IE2 and TRF2 or other telomeric proteins is complicated by our difficulty in immunoprecipitating the IE2 protein. To our knowledge, this is the first report identifying an HHV-6A/B protein localizing at cellular telomeres. Previous work from our laboratory identified IE2 as a Ubc9-interacting protein [61]. Considering that TRF2 can be sumoylated [62] and that Ubc9 is an E2 SUMO conjugating enzyme, it is conceivable that by localizing at telomere, IE2 through its interaction with Ubc9, may influence TRF2 SUMOylation. Knowing that SUMOylated TRF2 is prone to degradation [62], IE2 may therefore regulate TRF2 levels at chromosomal ends to facilitate integration and/or at VRC to free viral DNA from TRF2.

During infection, many viruses provoke a DNA damage response, either because their unprotected genome is recognized as damaged DNA or because of viral proteins triggering a damage signal. While several viruses have ways to evade the DDR pathways, some have developed strategies to make use of the cellular DNA repair proteins to their advantage. Cellular DNA repair proteins have been observed in VRC in various cases and can be helpful or even necessary for completion of the infection [63]. During EBV infection, the proteins involved in the ATM pathway checkpoint and HR repair are found in replication compartments [64]. The use of the DDR machinery by EBV likely increases the possibility of molecular events, stimulating the damage signals causing instability and promoting carcinogenic transformations. Whether viruses can use the DDR proteins in chromosomal integration is controversial, but some studies have suggested it [63]. One example is the Adeno-associated virus (AAV) that uses the cellular NHEJ mechanism for its site-specific integration [65]. HHV-6A/B chromosomal integration is not fully understood but it appears probable that these viruses integrate by HR between the virus' TMRs and the cellular telomeres. The integration occurs solely in telomeres and it has been shown that the telomeric sequences within the HHV-6A genome are essential for efficient integration into chromosomes [6]. Furthermore, the integrated virus has

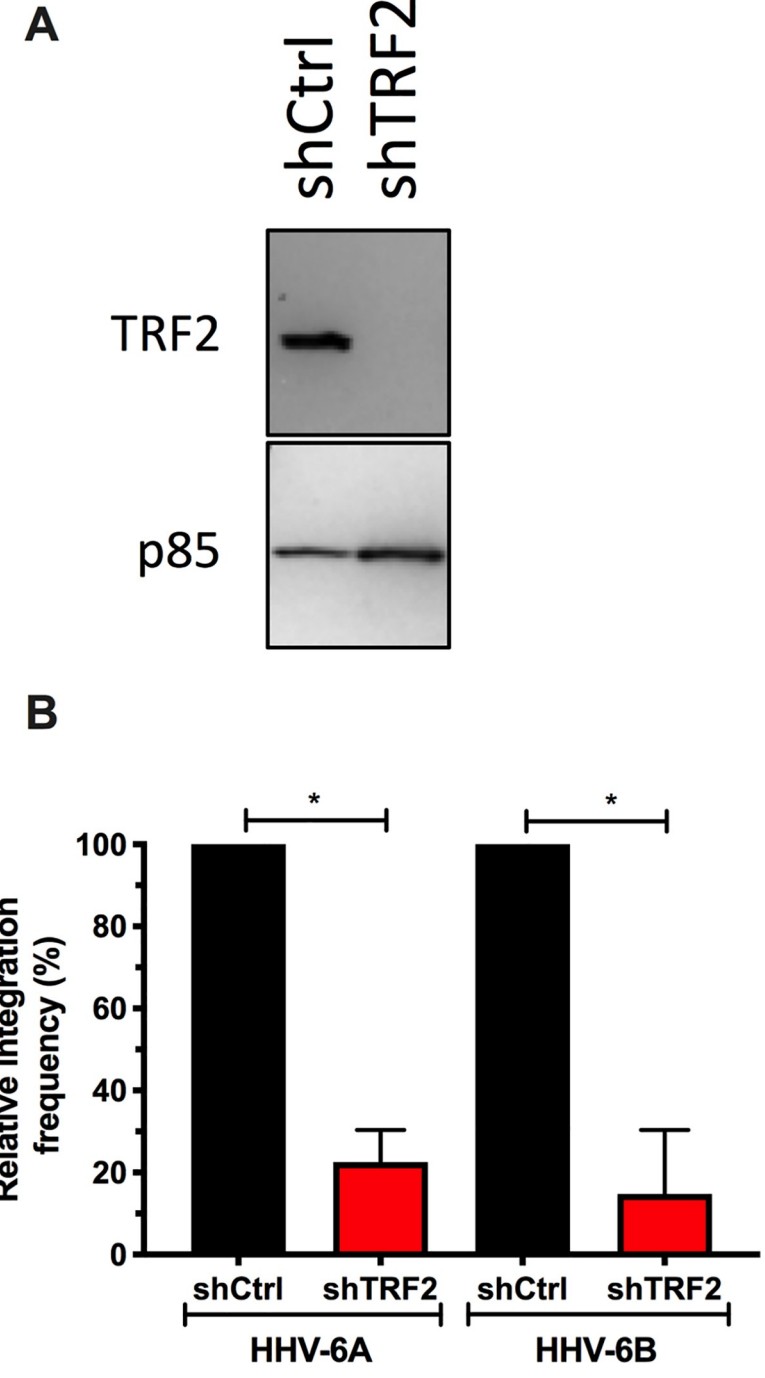

**Fig 8. TRF2 knockdown affect HHV-6A/B chromosomal integration.** A) U2OS cells were transduced with lentiviral vectors expressing a scrambles shRNA (shCtrl) or a shTRF2. After a week of selection, cells were monitored for TRF2 expression by western blot. B) After a week of selection, shCtrl and shTRF2 treated cells were infected with HHV-6A or HHV-6B. After 30 days, DNA was isolated and the relative frequency of integration, relative to shCtrl set at 100%, estimated by ddPCR. *p<0.05.

been sequenced and its orientation and missing sections are compatible with an integration by HR between the viral TMR and the telomere [13, 66–68]. The functions of TRF2 binding to HHV-6A/B viral DNA were studied in the context of HHV-6A/B integration. Our results

indicate that knockdown of TRF2 impairs the ability of HHV-6A/B to integrate the host chromosomes without affecting the initial phases of infection. At least two non-mutually hypotheses can explain this result. First, TRF2 is known to protect telomeres from a DDR [69]. By binding to the viral TMR, TRF2 may play a similar role by shielding the end of the viral genome from DDR. Second, TRF2 has been shown to participate in HR [70, 71]. We therefore surmise that the presence of TRF2 at viral DNA favors chromosomal integration by facilitating HR events between the host telomeres and viral telomeric sequences.

Our studies have focused mostly on TRF2. However, our results indicate that during infection, TRF1 also localizes with VRC. This is not unexpected considering that TRF1, alike TRF2, binds with high affinity to double-stranded DNA containing TTAGGG repeats [26]. Considering that both TRF1 and TRF2 are docking sites for other shelterin proteins [72], it is expected that Rap1, TIN2 and TPP1 are likely to be recruited at VRC during infection. Furthermore, considering a similar role for TRF1 and TRF2 in maintenance of telomere stability, we speculate that knockdown of TRF1 would affect HHV-6A/B integration alike our TRF2 knockdown experiments.

Our work has unraveled the potential importance of the TRF2 during infection and integration. However, our studies are not without limitations. First, the fact that TRF2 null cell are non-viable [29] prevents us from conducting chromosomal integration studies in the complete absence of TRF2. Considering that the integration assay lasts a month, our results likely reflect 30 days survival of cells expressing minimal amounts of TRF2 and containing integrated HHV-6A/B. At this point it therefore cannot be excluded with certainty that integration of HHV-6A/B in the absence of TRF2 can occur. Another limitation of our study is the reliance on immortalized cells to study integration processes. Considering that immortalized cells have telomere elongation mechanisms not typically observed in most primary cells, results from the use of primary cells could differ from those presented in the current study. Unfortunately, many of our protocols require prolonged culture periods, preventing us from using primary cells. Lastly, although our results suggest that shelterin proteins, such as TRF2, likely play a role in facilitating HHV-6A/B integration, the data provided does not prove it. Additional studies addressing the precise integration mechanism are needed before such a conclusion can be reached.

In summary, our studies indicate that during HHV-6A/B infection, the number of telomeric repeats increases significantly, as a result of the replicating viral DNA that contains many TMRs. In the presence of such abundant viral TMRs, TRF2 is recruited to VRCs where it colocalizes with the viral IE2 protein. Reciprocally, the IE2 protein efficiently localizes at cellular telomeres in the presence of TRF2. Lastly, knockdown of TRF2 negatively affected viral integration into host telomeres, highlighting potential new roles for TRF2, and likely IE2, during HHV-6A/B infection and integration.

## Acknowledgments

We acknowledge the Bioimaging platform of the Infectious Disease Research Centre, funded by an equipment and infrastructure grant from the Canadian Foundation Innovation (CFI). SGG and VC are recipients of fellowships from the Fonds de Recherche Québec-Santé.

## Author Contributions

**Conceptualization:** Shella Gilbert-Girard, Annie Gravel, Louis Flamand.

**Data curation:** Shella Gilbert-Girard, Annie Gravel, Vanessa Collin.

**Formal analysis:** Shella Gilbert-Girard, Annie Gravel, Vanessa Collin, Louis Flamand.

**Funding acquisition:** Louis Flamand.

**Investigation:** Annie Gravel, Louis Flamand.

**Methodology:** Shella Gilbert-Girard, Annie Gravel, Vanessa Collin, Darren J. Wight, Benedikt B. Kaufer, Eros Lazzerini-Denchi, Louis Flamand.

**Resources:** Darren J. Wight, Benedikt B. Kaufer, Louis Flamand.

**Supervision:** Annie Gravel, Louis Flamand.

**Writing – original draft:** Shella Gilbert-Girard, Louis Flamand.

**Writing – review & editing:** Shella Gilbert-Girard, Annie Gravel, Vanessa Collin, Darren J. Wight, Benedikt B. Kaufer, Eros Lazzerini-Denchi, Louis Flamand.

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
