## [Decision Letter · Decision Letter 0]

2 Oct 2019

Dear Dr. Flamand,

Thank you very much for submitting your manuscript "Recruitment of TRF2 at viral telomeres prevents DNA-damage response and facilitates human herpesvirus 6A/B chromosomal integration" (PPATHOGENS-D-19-01517) for review by PLOS Pathogens. Your manuscript was fully evaluated at the editorial level and by independent peer reviewers. The reviewers appreciated the attention to an important problem, but raised some substantial concerns about the manuscript as it currently stands. These issues must be addressed before we would be willing to consider a revised version of your study. We cannot, of course, promise publication at that time.

We therefore ask you to modify the manuscript according to the review recommendations before we can consider your manuscript for acceptance. Your revisions should address the specific points made by each reviewer.

(1) A letter containing a detailed list of your responses to the review comments and a description of the changes you have made in the manuscript. Please note while forming your response, if your article is accepted, you may have the opportunity to make the peer review history publicly available. The record will include editor decision letters (with reviews) and your responses to reviewer comments. If eligible, we will contact you to opt in or out.

(2) Two versions of the manuscript: one with either highlights or tracked changes denoting where the text has been changed; the other a clean version (uploaded as the manuscript file).

Additionally, to enhance the reproducibility of your results, PLOS recommends that you deposit your laboratory protocols in protocols.io, where a protocol can be assigned its own identifier (DOI) such that it can be cited independently in the future. For instructions see http://journals.plos.org/plospathogens/s/submission-guidelines#loc-materials-and-methods

We hope to receive your revised manuscript within 60 days. If you anticipate any delay in its return, we ask that you let us know the expected resubmission date by replying to this email. Revised manuscripts received beyond 60 days may require evaluation and peer review similar to that applied to newly submitted manuscripts.

[LINK]

Sincerely,

Philip E. Pellett, PhD

Guest Editor

PLOS Pathogens

Shou-Jiang Gao

Section Editor

PLOS Pathogens

Kasturi Haldar

Editor-in-Chief

PLOS Pathogens

orcid.org/0000-0001-5065-158X

Grant McFadden

Editor-in-Chief

PLOS Pathogens

orcid.org/0000-0002-2556-3526

Three reviewers have provided comments on this new submission of a previously reviewed manuscript. The two original reviewers acknowledged the considerable strides the authors have made in addressing issues raised in the initial review. Nonetheless, they have numerous questions and concerns, many of which relate to the description of the work. In addition, a third reviewer has raised a number of issues that need to be addressed.

I think this is important and worthwhile work, and I do not want the perfect to be the enemy of the good. While the work was criticized for its descriptive nature, if descriptive work is done well, it can provide a solid foundation for subsequent mechanistic studies. I am thus willing to forego the mechanistic studies requested by a reviewer.

The current paper does not go far enough in acknowledging the boundaries of robust interpretation available based on the data provided. A Results paragraph that begins, “Limitations of our study include, …” would improve the paper. The claim that the shelterin complex plays a role ("facilitates") in integration is plausible, but the data provided do not prove it.

The reviewers have identified numerous specific items for which the description, display, or interpretation need attention. Upon inspection, I agree with their comments.

I will look forward to seeing a revised version of the paper that addresses those concerns.

Reviewer's Responses to Questions

**Part I - Summary**

Reviewer #1: An interesting paper addressing HHV-6A/B chromosomal integration. Overall well written. It describes a significant increase in telomeric repeats upon HHV-6A/B replication and a colocalization of IE2 with the cellular telomeres. Conversely, cellular telomere binding proteins bind to viral telomeric repeats. Specifically, TRF2 is suggested to be important for replication and integration of the virus.

The concept of herpesviruses interfering with telomeric function is not a novel one, but understanding how HHV-6A/B does this may be of particularly interest given the ability of these viruses to integrate in the telomere region. The paper is predominantly descriptive. It would have been interesting to focus more on the mechanisms of how HHV-6A or HHV-6B integration occurs.

Reviewer #2: Flamand et al. begins to address the fascinating mechanism of HHV-6A/B telomere integration and the potential involvement of the shelterin complex protein TRF2. Several lines of evidence support TRF2 interaction with HHV-6A/B viral TMRs (IF-FISH, in vitro binding assay, ChIP) which is paired with an overall increase in telomere signal and TRF2 expression. Importantly, the second iteration of this manuscript begins to suggest a functional outcome of TRF2 depletion during viral integration.

Overall, the manuscript is well written and is of high importance to the HHV-6A/B and general herpes field. Comments are directed toward the clarity/resolution of immunofluorescence data and refinement of conclusions in regard to the ddPCR based “HHV-6A/B integration assay.”

Reviewer #3: The manuscript by Gilbert-Gerard et al. addresses interesting questions related to the role of mammalian telomeric and telomere-related sequences that are present at the termini of the genomes of human herpesviruses 6A and 6B (HHV-6A and HHV-6B), viruses whose genomes can integrate into telomeres of human chromosomes. In this work, the authors found that, as might be expected in cells containing thousands of viral genomes, telomeric abundance increases in infected cells. In addition, the enhanced telomeric signals colocalize with nuclear regions that contain the viral IE2 and P41 proteins, suggesting that these domains correspond to virus replication compartments (VRC). As demonstrated for other herpesviruses, VRC are sites where coordinated transcription, genome replication, and capsid assembly and filling occur in close proximity. Shelterin components localize to VRC and the shelterin protein TRF2 binds to the viral telomeric repeats. Knockdown of TRF2 in induces a DNA damage response that localizes to VRC. The authors suggest that, in addition to the role the shelterin complex plays in preventing induction of DNA damage responses localized to VRC, it also contributes to the ability of the virus to integrate into host chromosomes.

Studies of how highly prevalent viruses interact with machinery responsible for maintaining the integrity of host chromosomes are of interest and importance. The important observations here are the association of shelterin components with virus replication compartments, and demonstrating that knockdown of TRF2 in induces a DNA damage response that localizes to VRC. The conclusion that this plays a role in chromosomal integration, as opposed to survival of cells in which integration has occurred, is less robust.

**Part II – Major Issues: Key Experiments Required for Acceptance**

Reviewer #1: 1. It is argued that TRF2 is not required for IE2 localization at viral replication compartment. These experiments are based on a system with Dox-inducible shRNA, but unfortunately, these experiments lack certain controls of the Dox system. Dox may have a number of off-target effects, which are not controlled for in the experiments (e.g. Ahler et al., Doxycycline alters metabolism and proliferation of human cell lines. PLoS One, 2013).

To appreciate the data in fig.8, a control for shTRF2 is missing; that is, a control which is also treated with Dox.

In fig.8D it is argued that 53BP1 is located in foci in the absence of TRF2. It may be correct, but Fig.8A shows that TRF2 is indeed expressed in the presence of Dox, and since there is no staining for TRF2 in Fig. 8D it is difficult to draw this conclusion.

Also, there seems to be an inconsistency between the expression level of TRF2+Dox in Fig.8A and the level found by immunofluorescence in Fig.8B, which raises the issue of the kinetics of TRF2 expression during Dox treatment.

2. In the title the authors claim that recruitment of TRF2 prevents DNA-damage response, but the investigation of this response is rather superficial and does not provide much inside into what is actually going on. It is speculated in the discussion that integration occurs through HR. Nevertheless, the authors restrict their investigation to only examine the expression of a phosphorylated 53BP1 by immunofluorescence, although 53BP1 in association with RIF1 (not examined) are known for promoting NHEJ repair. Indeed 53BP1-RIF1 may antagonize BRCA1 (not examined) and HR. A lot is known about these pathways, and it should be feasible to investigate them in much greater details to provide a better understanding of how viral integration occurs and how HHV-6A/B interferes with the DNA-damage response.

Other points:

3. Fig. 3B: The exposure of the Alu-probed blot should be similar to the anti-TRF2 with (TTAGGG)3 probe to evaluate the data, in particular since nothing is provided about how linearity in the exposure interval is verified. One drawback of dot blots is the lack of size separation and thus an accumulated signal from everything that may bind the probe.

The HHV-6 DR6 probe apparently binds something in anti-Ig immunoprecipitated samples. If developed longer, it could look quite similar to anti-TRF2 IP. So how do the authors know that what is recognized in anti-TRF2 is not simply “non-specifically” precipitated DR or something completely different with cross-reactivity to the probe?

4. Presentation of the kinetics for accumulation of viral DNA copies would make fig. 1D more convincing. What was the MOI for these infections?

5. To appreciate the lack of colocalization of p41 with telomeres (fig.6G), the same 3D imaging as shown for fig. 6E should be provided.

Reviewer #2: -Fig. 9b, lines 371-381: When using ddPCR, is it possible to specifically measure the “relative HHV-6A/B integration frequency” among a population of U2OS cells that contain either the HHV-6A/B genome integrated or within viral replication compartments? It’s clear that TRF2 depletion leads to a reduction in the relative levels of HHV-6A/B viral genomes following infection of U2OS cells for one month, but is this a decrease in telomere integration or viral DNA replication? Providing quantitative analysis of FISH-IF (HHV-6A/B DNA and TTAGGG probe) on shSCR-U2OS and shTRF2-U2OS cells by counting the number of integration events within the total population of cells would be a potential means of addressing this question.

Reviewer #3: Major issues (I used downloaded TIF versions of the figures.):

1. An important, but readily addressable problem in the paper is that the figures and text do not explicitly state that the confocal microscopic images are of small numbers of nuclei, or of single nuclei. None of the images have scales. In the absence of DAPI staining (e.g., Fig. 1B) it would be helpful to provide outlines of nuclei. The DAPI images in Fig. 7 are too dim to be useful.

2. The Fig. 4B y-axes should be something like: “Number of HHV-6A genomes/Number of host genomes”. In Fig. 4C, the number of HHV-6B genomes per cell (~80,000/cell) is extraordinarily high, especially since only 2 of the 5 nuclei shown in Fig. 1C appear to be infected with HHV-6B. Please double-check the math.

3. In Fig. 5A, it would be helpful to show a merged image. It appears that the two images under IE2+ conditions are not presented at the same scale.

4. The explanation of the Fig. 9 integration frequencies is inadequate. A brief description of the method is needed (not just the reference). It is not clear whether in this context, the assay measures integration frequency, as opposed to the frequency of 30-day survival of cells in which integration occurred.

5. Why are there so many more IE2 spots in TRF1 and POT1 transfected cells than in non-transfected controls? IE2 is present in numerous donut-like structures in the transfected cells, but not in the non-transfected cells. How might this affect interpretations?

**Part III – Minor Issues: Editorial and Data Presentation Modifications**

Reviewer #1: 1. Fig. 8: It needs to be clearly specified when the western blotting for shRNA is performed and when the infection experiment and the immunofluorescence staining is performed to make sure the expression on the Western blot is representative for the expression at the time of immunofluorescence analyses.

2. The experimental conditions behind fig. 1E is not clearly described. Although some references to other papers are provided one should be able to understand the concepts from this paper alone. Thus, how was the data adjusted for the fact that uninfected cells keep dividing whereas infected cells stop dividing? How was telomeric repeats estimated?

3. l.61: …that play a role… -> that are required

4. l.315: “generally used” is an overstatement, as it seems only to refer to the authors themselves. Rephrase to “has been used”

5. l.505: mead -> mean

6. l.526: “uninfected (white)” is “uninfeceted (blue)” on my computer??

7. l.350: “colocalized perfectly with VRC, as evidenced by IE2 staining” It is not clear what qualifies it to be perfectly. Next, what is actually shown is colocalization to IE2. Therefore, it should be rephrased to “colocalized with IE2”

8. l.358, l.359 and several other places: It seems that the designation VRC is an interpretation. Better to describe what is actually examined and observed. In fig. 1G HHV-6A delta(Imp TMR) induce pronounced telomeric signals One colocalize very well with IE2, the other not very well – how should this be interpreted? And what does this mean for using IE2 staining as evidence of VRC?

9. Fig. 6C does does not provide additional information to what is already stated in the text. Thus, it is not necessary.

10. Fig. 8C: In my version of the paper, the mock cells and the HHV-6A cells -Dox (which are negative for antibody staining) have also much more dimly DAPI staining. Why is that?

(It may raise concerns about the compensation strategies during the collection of the immunofluorescent pictures.)

11. Fig.4: In knock-down experiments it is concluded that depletion of TRF2 does not affect viral replication. In particular because of the lack of an effect of the depletion, it is not clear whether the remaining amount of TRF2 is sufficient to exert a given function.

Without this, fig. 4 is difficult to interpret and does not add much.

Reviewer #2: Comments in this passage are focused primarily on the lack of clarity for some of the immunofluorescence images. It’s unknown whether the lack of clarity is attributed to low image resolution or inefficient staining.

-Figs. 1b, 6a, 6b, 6e: Including a DAPI counterstain would more clearly define the location of the cell given that its unknown whether these images contain one cell or multiple cells in the field of view.

-Fig. 1b, lines 232-236: Given the low image resolution, it’s difficult to differentiate between punctate (cellular) and “splotch” telomere signal. Would suggest providing a higher magnification of each example paired with localization of IE2 to more clearly convey this point.

-Fig. 6a: Surprising the telomere signal is very low (nearly absent) in mock group for Fig. 6a, but not in Fig. 6b. Furthermore, unable to observe “IE2 is also perfectly colocalized with VRC alongside p41” again due to low fluorescence intensity. May also want to include a comment in text describing the highlighted area containing the white box.

-Figs. 6b, 6f: How many cells were counted? Information is missing from the text.

-Figs. 7a, 7b: IE2 signal is very low in “HHV-6A; non-transfected” group. POT1 signal is also very low in the “POT1 transfected” group.

-Fig. 8d: In contrast to “HHV-6A (+) DOX” group, the “patchy” TTAGGG signal is absent from regions of IE2 in the “HHV-6A (-) DOX” group. Moreover, it appears that TTAGGG punctate foci are excluded from the “more pronounced VRC.” This observation appears to be different from other figures in this manuscript.

-Lines 194-95, Fig. 3: Unclear why an input sample was taken prior to sonication. Input should have been from the same pool of chromatin that was used in the TRF2 and IgG ChIP. It’s known that sonication aids in cell lysis during chromatin shearing, therefore the % input used in the figure does not accurately represent the efficiency of the ChIP.

-Line 199, Fig. 3: State the quantity (µg) of normal rabbit IgG used in the ChIP (not volume). If the same µg of antibody was not used for IgG and TRF2, authors would need to specify reason when representing background signal.

-Figs. 3c, 3e: If results are from 3 independent experiments, error bars and % input for IgG (background signal) is missing. Including IgG in this figure is critical given the noticeable IgG signal in HHV-6A and HHV-6B infected cells (see Figs. 3b and 3d).

-Line 206, Fig. 2c: Why were different amounts of MBP (25 ng) and MBP-TRF2 (50 ng) used in the in vitro binding assay? Puzzling given that MBP alone serves as a control for the nonspecific binding to HHV-6A DNA.

-Line 295-296: Consider including RNAPII ChIP data into Fig. 3 or supplementary information rather than stating data is “not shown.”

-Fig. 8a: Would suggest re-running this western to achieve individual clean bands with densitometry analysis.

-Given the similar roles of TRF1 and TRF2 in the maintenance of telomere stability, may want to include a very brief comment why TRF2 and not TRF1 was the primary focus? Could there be redundancy between both TRF1 and TRF2, such that depleting both proteins or TRF1 alone leads to a greater decrease in viral integration? This is purely speculative…experiments are not necessary.

Reviewer #3: Suggested edits or points needing clarifications:

1E – y-axis needs a label – is this intensity or copy number

Fig. 2. Panels A and B should have competitors expressed in pmoles, as is done for Panel C.

In Panel A, what are the units for the 10, 100, and 1000 telo and non-telo numbers

In Panel C, what are the molar ratios between the target (25 ng of genomes) and the competitors?

Fig. 4A. It is hard to tell whether the bands seen in the +Dox lane are background expression or leakage from the adjacent lane. If it is the latter, the authors’ case would be stronger.

Silencing TRF2 seems to have a small but fairly consistent positive effect on HHV-6A genome replication.

Fig. 6A. No scales are provided. Are these high-magnification images of single nuclei? State that the boxes are the region shown in the 3D representation. What do you make of the IE rings (more common with the deletion mutant)? The difference in localization relative to the TTAGGG probe should be mentioned.

In 6C, it is easy to see the relationship of the patterns for the TTAGGG, IE2, Merged, and XYZ panels, because they are presented in the same relative positions in all four panels. For the IE2 deletion data in panel 6E, the XYZ panel is rotated relative to the others. I do not see how the XYZ panel in the empty vector row relates to the first three panels. Related to this, I could only figure out how the deletion 3D image relates to the rest in its row. What is the scale? What is the definition of “at telomere”? It would be helpful to show at least two images of the 3D reconstructions: one in an orientation that correspoinds to the XYZ panel (which should correspond to the others in that row), and one rotated to make whatever other point is intended. Movies that show the reconstruction rotating in space would be helpful.

The Fig. 6G legend text should be modified to say “Lack of colocalization between HHV-6A p41 and telomeres in uninfected cells.”

Fig. 7A and 7B. In the top rows, why are there so many more IE2+ spots in the 3D rendering?

Fig. 8B. Why are the TTAGGG signals so much more pixelated than the other signals (looking at the high resolution TIFF images)?

Why is the IE2 staining in Fig. 8 panel B so much more diffuse than in Panels C and D. This is not an image size or resolution issue.

Fig. 9B y-axis: “frequency”

line 505. “mean absorbance”

PLOS authors have the option to publish the peer review history of their article (what does this mean?). If published, this will include your full peer review and any attached files.

Reviewer #1: No

Reviewer #2: No

Reviewer #3: No

---

## [Decision Letter · Decision Letter 1]

11 Mar 2020

Dear Dr. Flamand,

Thank you very much for submitting your manuscript "Role for the shelterin protein TRF2 in human herpesvirus 6A/B chromosomal integration" for consideration at PLOS Pathogens. As with all papers reviewed by the journal, your manuscript was reviewed by members of the editorial board and by several independent reviewers. The reviewers appreciated the attention to an important topic. Based on the reviews, we are likely to accept this manuscript for publication, providing that you modify the manuscript according to the review recommendations.

The Reviewers have given this paper very careful and thorough consideration. They find the work an interesting and important contribution to understanding the roles of TRF2 in HHV-6 chromosomal integration. Reviewer 1 suggests correction of what is essentially a typo. Reviewer 2 has a handful of reasonable questions and suggestions related to figures and the image analysis; I would like to see a revision that addresses their concerns.

Sincerely,

Philip E. Pellett, PhD

Guest Editor

PLOS Pathogens

Shou-Jiang Gao

Section Editor

PLOS Pathogens

Kasturi Haldar

Editor-in-Chief

PLOS Pathogens

orcid.org/0000-0001-5065-158X

Michael Malim

Editor-in-Chief

PLOS Pathogens

orcid.org/0000-0002-7699-2064

The Reviewers have given this paper very careful and thorough consideration. They find the work an interesting and important contribution to understanding the roles of TRF2 in HHV-6 chromosomal integration. Reviewer 1 suggests correction of what is essentially a typo. Reviewer 2 has a handful of reasonable questions and suggestions related to figures and the image analysis; I would like to see a revision that addresses their concerns.

Reviewer Comments (if any, and for reference):

Reviewer's Responses to Questions

**Part I - Summary**

Reviewer #1: HHV-6A/B can integrate in host cell chromosomes, but the mechanism is unknown. The revised version of this paper has been significantly improved by addressing some critical issues and by presenting the data in a more focused way.

The authors describe that productive HHV-6A/B infections increase viral telomeric sequences; shelterin proteins are recruited to the viral replication compartment; shelterin protein TRF2 binds to viral telomeric repeats; TRF2 knockdown reduces localization of viral IE2 to cellular telomeres and reduces frequency of viral integration. They conclude that TRF2 has a role in HHV-6/B integration.

Although the paper does not provide a mechanism for the chromosomal integration process, it does provide a first step in the understanding of this apparently unique feature of HHV-6A/B among the human herpesviruses.

Reviewer #2: With this current submission the authors have addressed a number of questions that were brought up during the course of review. Comments are provided to aid in the clarity in some of the results and conclusions.

**Part II – Major Issues: Key Experiments Required for Acceptance**

Reviewer #1: (No Response)

Reviewer #2: -Fig 1b, lines 247-249: Based on the images provided it appears that strong IE2 signal is excluded from the p41 signal, hence the two proteins are adjacent rather than colocalizing. What is the Mander’s colocalization coefficient for p41 and IE2? This is important to establish given the use of IE2 as a marker for viral replication compartment in all subsequent figures. Also, poor image resolution (pixilated) may be a culprit for the lack in clarity.

**Part III – Minor Issues: Editorial and Data Presentation Modifications**

Reviewer #1: l.35: ”integration frequency” is mentioned twice in the same sentence. Please correct.

Reviewer #2: -Fig 1b, lines 251-253: Based on very low signal “telomeric probe resulted in the detection of many discrete punctate telomeric signal” is not apparent in Fig 1b, but is in Fig 1c. Again, this is the same general issue as brought up in the previous review of low signal or low image resolution.

-Fig 1g, lines 275-284: In this figure it is obvious that HHV-6A and HHV-6AdeltaimpTMR have an IE2 signal that localizes with telomere. However, without providing a positive IE2 infected cell for HHV-6AdeltaTMR its difficult to conclude “the viral genome were responsible for increased telomeric signals observed.” To make this conclusion an HHV-6AdeltaTMR infected cell with positive IE2 signal would need to be shown with telomere signal that does not colocalize. Poor image resolution maybe a culprit again, but the data presented should be clear.

-Fig 4a, lines 335-336: Provide the Mander’s colocalization coefficient to conclude “TRF2 appeared to colocalize with IE2.”

-Fig 5a, lines 564-566: Authors need to define the difference between patchy and punctate IE2 signal with regards to VRC. Is it possible to conclude that “infected cells that do not actively replicate viral DNA with TRF2 and IE2 colocalization” when there are no viral DNA FISH controls to support these conclusions of which IE2 staining pattern (patchy vs punctate) represents VRCs?

-Fig 5b, lines 346-351: The % colocalization of punctate IE2 with TRF2 was provided in Fig 5a, but this information is missing for IE2 and TRF1.

-Fig 6d, lines 363-365: The p41 signal appears to be localized as a concentric ring around the nucleus and not “p41 exhibit a diffuse nuclear distribution.”

-Lines 589-590, Fig 7c: Describe how the percent of IE2 positive cells were “estimated” as opposed to providing an absolute count instead?

PLOS authors have the option to publish the peer review history of their article (what does this mean?). If published, this will include your full peer review and any attached files.

Reviewer #1: No

Reviewer #2: No
---

## [Editor Report · Decision Letter 2]

24 Mar 2020

Dear Dr. Flamand,

Thank you for your positive responses to the reviewers throughout this process. We are very pleased to inform you that your manuscript 'Role for the shelterin protein TRF2 in human herpesvirus 6A/B chromosomal integration' has been provisionally accepted for publication in PLOS Pathogens.

Best regards,

Philip E. Pellett, PhD

Guest Editor

PLOS Pathogens

Shou-Jiang Gao

Section Editor

PLOS Pathogens

Kasturi Haldar

Editor-in-Chief

PLOS Pathogens

orcid.org/0000-0001-5065-158X

Michael Malim

Editor-in-Chief

PLOS Pathogens

orcid.org/0000-0002-7699-2064
---

## [Editor Report · Acceptance letter]

15 Apr 2020

Dear Dr. Flamand,

We are delighted to inform you that your manuscript, "Role for the shelterin protein TRF2 in human herpesvirus 6A/B chromosomal integration," has been formally accepted for publication in PLOS Pathogens.

Best regards,

Kasturi Haldar

Editor-in-Chief

PLOS Pathogens

orcid.org/0000-0001-5065-158X

Michael Malim

Editor-in-Chief

PLOS Pathogens

orcid.org/0000-0002-7699-2064